# Integrated sensing of host stresses by inhibition of a cytoplasmic two-component system controls *M. tuberculosis* acute lung infection

John A Buglino[1], Gaurav D Sankhe[1], Nathaniel Lazar[2], James M Bean[1], Michael S Glickman[1,2,3]*

[1]Immunology Program Sloan Kettering Institute, New York City, United States; [2]Immunology and Microbial Pathogenesis Graduate Program, Weill Cornell Graduate School, New York City, United States; [3]Division of Infectious Diseases, Memorial Sloan Kettering Cancer Center, New York City, United States

**Abstract** Bacterial pathogens that infect phagocytic cells must deploy mechanisms that sense and neutralize host microbicidal effectors. For *Mycobacterium tuberculosis*, the causative agent of tuberculosis, these mechanisms allow the bacterium to rapidly adapt from aerosol transmission to initial growth in the lung alveolar macrophage. Here, we identify a branched signaling circuit in *M. tuberculosis* that controls growth in the lung through integrated direct sensing of copper ions and nitric oxide by coupled activity of the Rip1 intramembrane protease and the PdtaS/R two-component system. This circuit uses a two-signal mechanism to inactivate the PdtaS/PdtaR two-component system, which constitutively represses virulence gene expression. Cu and NO inhibit the PdtaS sensor kinase through a dicysteine motif in the N-terminal GAF domain. The NO arm of the pathway is further controlled by sequestration of the PdtaR RNA binding response regulator by an NO-induced small RNA, controlled by the Rip1 intramembrane protease. This coupled Rip1/ PdtaS/PdtaR circuit controls NO resistance and acute lung infection in mice by relieving PdtaS/R-mediated repression of isonitrile chalkophore biosynthesis. These studies identify an integrated mechanism by which *M. tuberculosis* senses and resists macrophage chemical effectors to achieve pathogenesis

**\*For correspondence:** glickmam@mskcc.org

## Introduction

Intracellular bacterial pathogens such as *Mycobacterium tuberculosis* and Salmonella that infect macrophages have evolved elaborate mechanisms to respond to the toxic environment of the macrophage phagosome and associated endocytic compartments (*Olive and Sassetti, 2016*; *Stallings and Glickman, 2019*). Phagosomal pathogens must contend with such host-inflicted stresses as oxidative stress (*Bustamante et al., 2011*), nitrosative stress (*Darwin et al., 2003*; *Fang and Vázquez-Torres, 2019*), iron deprivation, copper and zinc toxicity (*Botella et al., 2011*; *Sheldon and Skaar, 2019*; *Shi and Darwin, 2015*), and low pH (*Vandal et al., 2009*), among others. The pathogen response to these molecules is multifaceted and must be dynamic and graded to respond both to rapidly changing environments and the prospect of combinations of stresses that vary in intensity and composition over time. When expelled by coughing and inhaled by the naïve host, *M. tuberculosis* must transition from the nutrient-rich environment of the pulmonary cavity to deposition in the alveolus and engulfment by an alveolar macrophage. Such rapid transitions in environment require rapid changes in gene expression for successful adaptation.

Bacterial two-component systems (TCS) are widespread sensing systems that respond to a wide variety of ligands, including ions, gases, and metabolites. For pathogenic bacteria, TCS promote pathogenesis by modifying bacterial gene expression in response to host-inflicted toxic stresses or metabolic environments (*Bretl et al., 2011*; *Groisman, 2016*). The classic TCS sensing system consists of a membrane-bound sensor kinase that senses an extracellular ligand and activates through autophosphorylation on a cytoplasmic histidine. Transfer of this phosphate to an aspartate in the receiver domain of the cognate response regulator (RR) activates the RR to bind DNA of its target genes, thereby controlling gene expression (*Sankhe et al., 2018*; *Zschiedrich et al., 2016*).

In addition to TCS signaling, proteolysis is another widespread mechanism of bacterial signal transduction in which membrane-embedded proteases process membrane-embedded proteins, often anti-sigma factors (*Schneider and Glickman, 2013*; *Urban, 2009*). The S2P class of intramembrane proteases is widely distributed in bacteria, and several have been implicated in controlling virulence functions of bacterial and fungal pathogens, including *Vibrio cholerae* (*Almagro-Moreno et al., 2015*; *Matson and DiRita, 2005*), Cryptococcus (*Bien et al., 2009*), and *M. tuberculosis*. *M. tuberculosis* Rip1 is an important virulence determinant required for both acute growth in the lung and long-term persistence during chronic infection (*Makinoshima and Glickman, 2005*). Rip1 controls four independent sigma factor pathways through four anti-sigma factors substrates (*Schneider et al., 2014*; *Sklar et al., 2010*), but the virulence function of Rip1 appears to be independent of these pathways (*Sklar et al., 2010*). Given the importance of this pathway for *M. tuberculosis* pathogenesis, we sought to determine the virulence pathway(s) controlled by Rip1. These investigations, described below, uncover a new signaling system that integrates the bacterial response to Cu and NO and thereby controls growth in the host lung. The hub of this signaling system is the PdtaS/R cytoplasmic TCS, which constitutively represses virulence gene expression until inactivation by Cu and NO. The NO arm of the pathway is further controlled by titration of PdtaR from its RNA targets by an NO-induced, Rip1-controlled, small RNA, which binds directly to PdtaR and controls expression of isonitrile chalkophores. The ultimate Cu resistance mechanism controlled by Rip1/PdtaS/R is independent of chalkophores and remains to be identified.

## Results

### The Rip1 pathway defends against metal and nitrosative stress

To understand the basis for the severe virulence defect of *M. tuberculosis* lacking the Rip1 protease (*Makinoshima and Glickman, 2005*), a phenotype that is independent of the four identified Rip1-controlled sigma factor pathways (*Schneider et al., 2014*; *Sklar et al., 2010*), we hypothesized that Rip1 defends against specific host-imposed bactericidal effector molecules. We performed bacterial killing assays of wild-type (WT) *M. tuberculosis* and Δ*rip1* with nitric oxide, hydrogen peroxide, copper, zinc, low pH, detergent, lysozyme, and nutrient starvation. We observed no Rip1-dependent sensitization to starvation, lysozyme, oxidative stress, detergent, or low pH (*Figure 1—figure supplement 1A–C*). However, we observed that Rip1 is required for resistance to copper ions, nitric oxide, and zinc. M. tuberculosis Δ*rip1* is 10,000-fold more sensitive to copper on agar media than wild-type Mtb, and this phenotype is restored by genetic complementation by a plasmid encoding a wild-type copy of Rip1, but not a proteolytically inactive Rip1 (*Figure 1A*). Sensitivity was also observed for Δ*rip1* cells grown in liquid media with Cu supplementation (*Figure 1—figure supplement 1D*). Similarly, Δ*rip1* growth is inhibited by 100 μM Zn (*Figure 1B*), but not iron (*Figure 1C*). Rip1 is also required for resistance to nitric oxide. Treatment of wild-type Mtb for 3 days with 200 μM of the NO donor diethylenetriamine nitric oxide (DETA-NO) minimally reduced bacterial viability, whereas Δ*rip1* titers were reduced 100-fold, a phenotype that was also complemented with wild-type Rip1 but not proteolytically inactive Rip1 (*Figure 1D*). Importantly, similar metal nor NO sensitivity was observed for M. tuberculosis strains lacking individual or any tested combination of Rip1-controlled sigma factor pathways (*Figure 1—figure supplement 2A,B*), indicating that the Rip1-controlled metal/NO resistance pathway is a previously unrecognized, sigma factor-independent arm of the Rip1 pathway.

Multiple prior studies have investigated copper and NO response and resistance mechanisms in *M. tuberculosis*. Copper is sensed by the metal binding repressors RicR and CsoR (*Festa et al., 2011*; *Liu et al., 2007*; *Marcus et al., 2016*; *Shi and Darwin, 2015*), which control regulons involved

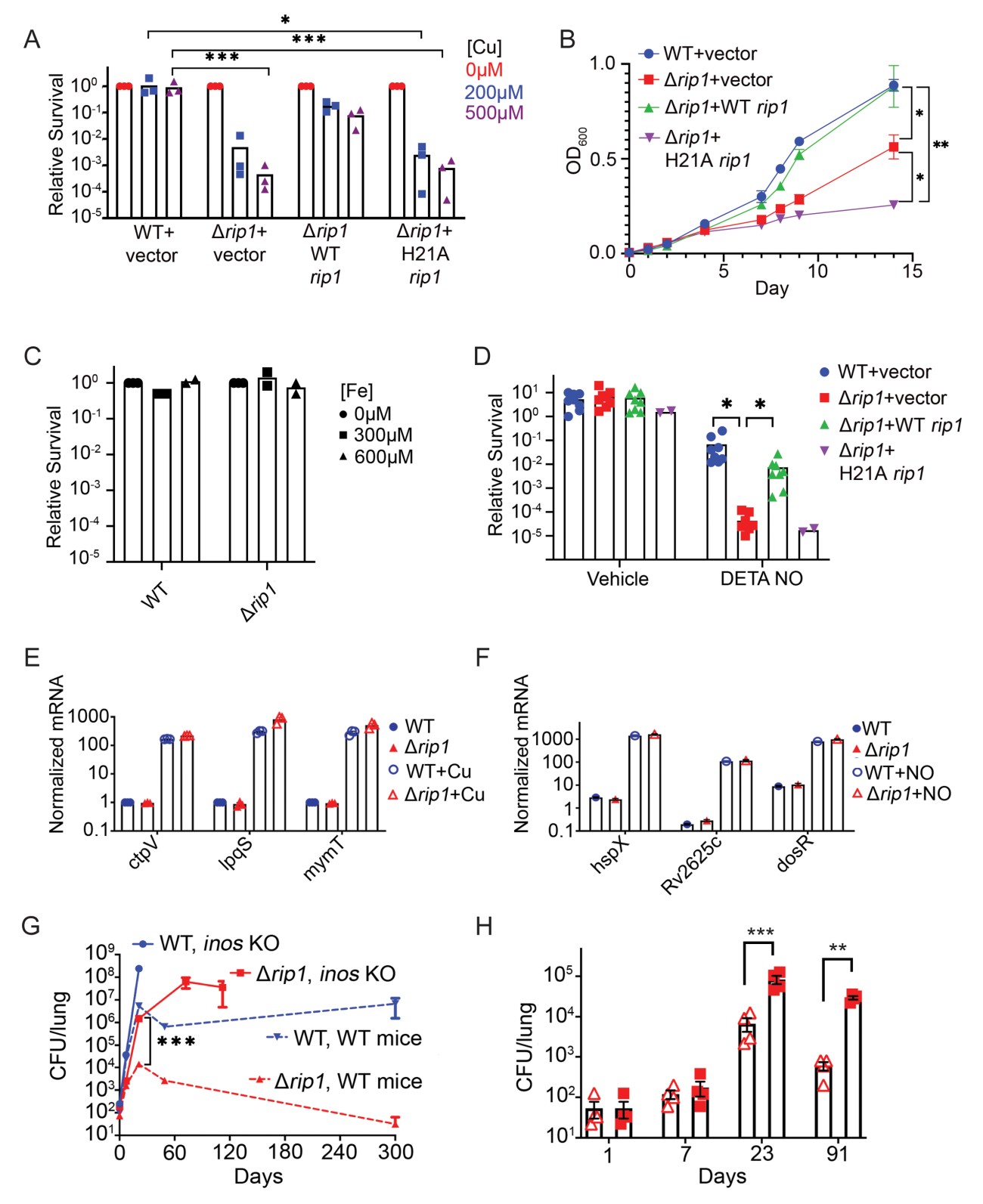

**Figure 1.** The Rip1 protease controls copper and nitric oxide resistance. (**A**) The Rip1 pathway confers resistance to copper. Bacterial colony-forming units (CFUs) of the indicated strains grown on agar plates supplemented with 200 or 500 µM copper sulfate. Relative survival is normalized to untreated controls, which are set at $10^0 = 1$. Each value is the average of technical duplicate measurements for n = 3 biological replicates. Statistical analysis by two-way ANOVA with Tukey's multi-comparison correction *p<0.01 WT + vector 200 µM vs. $\Delta rip1$ +H21A $rip1$; ***p<0.001 WT + vector 500 µM vs.

*Figure 1 continued on next page*

Figure 1 continued

Δ*rip1* +vector; WT + vector 500 μM vs. Δ*rip1* +H21A *rip1*. (B) The Rip1 pathway confers resistance to zinc. Growth (OD$_{600}$) of the indicated strains in liquid culture supplemented with 100 μM zinc sulfate. Data plotted is SEM of n = 3 biological replicates. Statistical analysis by ordinary one-way ANOVA with Tukey's multi-comparison correction. *p<0.05, **p<0.01; day 14 WT + vector vs. Δ*rip1* + vector p=0.027; day 14 Δ*rip1* +vector vs. Δ*rip1* + H21A *rip1* p=0.032; day 14 WT + vector vs. Δ*rip1* + H21A *rip1* p=0.002. (C) The Rip1 pathway does not control iron sensitivity. Growth on agar plates supplemented with 300 or 600 μM iron chloride normalized to untreated controls as in (A). Each value is the average of technical duplicate measurements for n = 2 biological replicates. (D) The Rip1 pathway confers resistance to nitric oxide. CFU counts of the indicated strains post treatment with vehicle or 200 μM diethylenetriamine nitric oxide adduct (DETA-NO) for 3 days. Relative survival = CFU day 3 post treatment/CFU at day 0 for each condition n = 8 biological replicates for WT + vector, Δ*rip1* + vector, and Δ*rip1* + WT rip1, and n = 2 for Δ*rip1* + H21A *rip1*. Statistical analysis by unpaired t-test. *p<0.05; DETA-NO WT + vector vs. Δ*rip1* + vector p=0.041; DETA-NO Δ*rip1* + vector vs. Δ*rip1* + WT *rip1* p=0.031. (E) Loss of Rip1 does not affect Cu-induced transcription. RT-qPCR quantitation of known copper-regulated transcripts *ctpV*, *lpqS*, and *mymT* between WT (blue) and Δ*rip1* (red) cells in resting (filled symbols) or 200 μM copper sulfate (open symbols) conditions for 2 hr. Values are normalized to *sigA* transcript levels. Values represent average of three technical replicate measurements of three biological replicates. (F) Loss of Rip1 does not affect nitric oxide-regulated gene expression through DosR. RT-qPCR comparison of nitric oxide-induced, DosR-regulated transcripts *hspX*, *rv2625c*, and *dosR* between WT and Δ*rip1* cells following 3 hr of treatment with vehicle (filled symbols) or 200 μM DETA-NO (empty symbols). Values are normalized to *sigA* transcript levels. Mean of biological triplicates reported. (G) NO mediates attenuation of *M. tuberculosis* Δ*rip1* in mouse lung infection. Bacterial burden (CFU) in lungs of C57bl/6 (broken lines), and age-matched *Nos2-/-* (solid lines) mice infected with WT *M. tuberculosis* (blue) or *M. tuberculosis* Δ*rip1* (red) at the indicated times (days) post infection. Data plotted is mean and SD of n = 4 biological replicates. ***p=0.0003 by unpaired t-test at 21-day time point for Δ*rip1* in WT vs. Nos2 -/-. (H) Repeat experiment as in (G), but with lower initial inoculum of Δ*rip1* *M. tuberculosis* in WT (open triangles) and Nos2-/- (closed squares) mice. For all strains/host day 1 n = 3 biological replicates; n = 4 biological replicates for all other time points. Statistical significance by unpaired t-test analysis is represented as *p<0.05, **p<0.01, ***p<0.001.

The online version of this article includes the following figure supplement(s) for figure 1:

**Figure supplement 1.** Rip1 does not defend against starvation, oxidative stress, detergent, or acid.

**Figure supplement 2.** Rip1-controlled sigma factor pathways do not contribute to Cu or NO resistance.

**Figure supplement 3.** Rip1 copper sensitivity is not through known Cu resistance pathways.

in copper binding and export (*Rowland and Niederweis, 2012*; *Shi et al., 2014*; *Ward et al., 2010*). We tested whether relief of RicR or CsoR repression was impaired in Δ*rip1* by measuring copper-induced transcription of their target genes. We found no defect in the copper-induced transcription of *mymT* or *lpqS* (RicR targets; *Festa et al., 2011*) or CtpV (CsoR target) in Δ*rip1* cells, indicating functionality of these systems (*Figure 1E*, and *Figure 1—figure supplement 3A,B*). To determine whether a Rip1-dependent post-translational modification of copper export or chelation by CtpV (*Ward et al., 2010*) or MymT (*Gold et al., 2008*) might explain the copper sensitivity of Rip1, we performed genetic epistasis tests in Δ*rip1*/Δ*ctpV* and Δ*rip1*/Δ*mymT*. We found minimal copper sensitivity of Δ*ctpV* or Δ*mymT*, and no enhanced copper sensitivity of Δ*rip*/Δ*ctpV* or Δ*rip1*Δ*mymT* compared to Δ*rip1* (*Figure 1—figure supplement 3C*). Additionally, de-repression of the CsoR regulon in the Δ*rip1* background (through Δ*rip1*Δ*csoR*) had no effect on Δ*rip1* copper sensitivity (*Figure 1—figure supplement 3C*). These results indicate that the Rip1 copper sensitivity is not due to known pathways of copper chelation or efflux. We similarly examined known NO response regulons, but found that NO-induced transcription of three DosS/T/R target genes, which controls the dominant transcriptional response to NO (*Voskuil et al., 2003*), was intact in Δ*rip1*, indicating intact NO sensing at the cell surface through this TCS (*Figure 1F*). Our results indicate that Rip1 controls a pathway of resistance to Cu and NO, which is likely distinct from previously defined pathways.

## Nitric oxide-dependent attenuation of *M. tuberculosis* Δ*rip1*

To test the importance of the Rip1-dependent NO resistance pathway in vivo, we infected NOS2-deficient mice. We reasoned that if NO is a significant attenuating pressure for *M. tuberculosis* Δ*rip1* in vivo, we would observe some reversal of the Δ*rip1* virulence defect when NO is removed, as has been reported for other NO-sensitive mutants (*Darwin et al., 2003*; *Darwin and Nathan, 2005*). NOS2-deficient mice infected with wild-type Mtb were highly susceptible to Mtb infection, as previously reported (*MacMicking et al., 1997*; *Figure 1G*). The impaired growth of *M. tuberculosis* Δ*rip1* in the lung was dramatically reversed in the absence of NO, such that Δ*rip1* lung titers in NOS2-deficient animals were 100-fold higher than in wild-type mice at 3 weeks and 10,000-fold higher at 8–10 weeks post infection (*Figure 1G*). A repeat infection at a lower inoculum confirmed these findings (*Figure 1H*). These data establish that the Rip1 pathway defends against NO in vivo during acute lung infection, possibly through a direct antimicrobial effect of NO.

## The PdtaS/PdtaR TCS controls NO and Cu resistance downstream of Rip1

The data presented above indicate that the Rip1 protease controls a pathway that defends against Cu and NO, but that this phenotype is neither attributable to previously identified Rip1-controlled sigma factor pathways nor due to known mechanisms of Cu or NO sensing. To examine the molecular basis for these phenotypes, we executed a genetic suppressor screen for spontaneous chromosomal mutations that revert the Cu sensitivity phenotype of *M. tuberculosis* Δ*rip1* (*Figure 2A*). *M. tuberculosis* Δ*rip1* was selected on agar media containing 500 μM CuSO₄, and surviving bacteria were clonally purified, and after confirming the acquisition of Cu resistance, examined by whole genome sequencing (*Figure 2A*). We detected two suppressor strains with distinct mutations in the *rv3220* gene, encoding the PdtaS sensor kinase: a frameshift mutation at amino acid 37 of the encoded protein (F37X) and a substitution mutation V54F. Both mutations were confirmed by sequencing of chromosomal segments amplified by PCR from the suppressor strains (*Figure 2—figure supplement 1A,B*). In contrast to most sensor kinases that are membrane bound, PdtaS is a soluble sensor kinase with a C-terminal histidine kinase domain and N-terminal sensing GAF (in which the suppressor mutations were detected) and PAS domains (*Figure 2B*). Its structure is known (*Preu et al., 2012*), but its function is not understood.

To confirm that the PdtaS mutations are the functional variants reverting the Cu sensitivity, we performed complementation tests on the suppressor strains with PdtaS or PdtaS variants predicted to lack kinase activity (*Figure 2C*). Introduction of PdtaS into either Δ*rip1pdtaS*(F37X) or Δ*rip1pdtaS* (V54F) restored Cu sensitivity to both suppressor strains, indicating that the PdtaS single nucleotide polymorphisms (SNPs) are the functionally important suppressor mutations (*Figure 2D*). In addition, we constructed a chromosomal deletion of *pdtaS* (Δ*rip1*Δ*pdtaS*) and observed the same pattern of PdtaS-dependent Cu sensitivity (*Figure 2D*). To probe the requirement of PdtaS kinase activity, we mutated histidines 302/303 to glutamines, and glycines 443/445 to alanine, which are required for autophosphorylation and kinase activity respectively (*Trajtenberg et al., 2010*), and expressed these variants in the Δ*rip1pdtaS*(F37X) strain. Although both kinase dead proteins were expressed to equivalent levels (*Figure 2C*), neither were able to restore Cu sensitivity (*Figure 2D*), indicating a requirement for PdtaS kinase activity in controlling Cu resistance.

PdtaS was reported to phosphorylate the RR PdtaR (*Morth et al., 2005*), which is itself an atypical RR in that it contains an ANTAR RNA binding domain (*Ramesh et al., 2012*) instead of the DNA binding domain present in many RRs. To validate the participation of PdtaR in the Δ*rip1* Cu sensitivity, we constructed Δ*rip1*Δ*pdtaR* and tested Cu sensitivity. We observed that loss of *pdtaR* reverted the Cu sensitivity (*Figure 2E*) of Δ*rip1*. Complementation of Δ*rip1*Δ*pdtaR* with *pdtaR* restored copper sensitivity (*Figure 2E*). However, expression of a PdtaR-D65A, which lacks the receiver aspartate phosphorylated by PdtaS, was partially active, partially restoring Cu sensitivity to Δ*rip1*Δ*pdtaR* at low-dose Cu and completely at higher-dose Cu (*Figure 2E*). These results suggest that phosphorylation of PdtaR is not absolutely required for its activity in the Cu sensitivity pathway.

The suppressor mutations in PdtaS were identified as suppressors of Cu sensitivity, but the NO sensitivity might be due to a different Rip1-dependent pathway. However, we observed that loss of *pdtaS* in Δ*rip1* also reverted the NO sensitivity nearly to the level of WT *M. tuberculosis* (*Figure 2F*), with the same requirement for PdtaS kinase activity. In addition, inactivation of *pdtaS* in wild-type cells conferred a hyperresistance to NO (*Figure 2F*). We next tested the involvement of PdtaR in the NO sensitivity phenotype of Δ*rip1* and found the loss of *pdtaR* reverted to the level of the wild-type strain (*Figure 2G*). We also observed a partial activity of PdtaR D65A, which partially complemented in comparison to WT PdtaR (*Figure 2G*). Loss of PdtaR in wild-type Mtb conferred hyperresistance to NO compared to the parental wild-type strain, again indicating that PdtaR suppresses NO resistance (*Figure 2G*). Taken together, these results demonstrate that the Cu and NO sensitivity conferred by loss of *rip1* proceeds through an active PdtaS/R signaling system and that these two sensitivity phenotypes represent an integrated signaling pathway that responds to both stresses.

## NO and Cu directly inhibit PdtaS kinase activity

The data above suggests that PdtaS/PdtaR acts as a negative regulator of Cu and NO resistance, and that this negative regulation is not relieved in the Δ*rip1* background. Genetic deletion of *pdtaS/R* relieves this inhibition and restores wild-type stress resistance. Although the PdtaS GAF domain

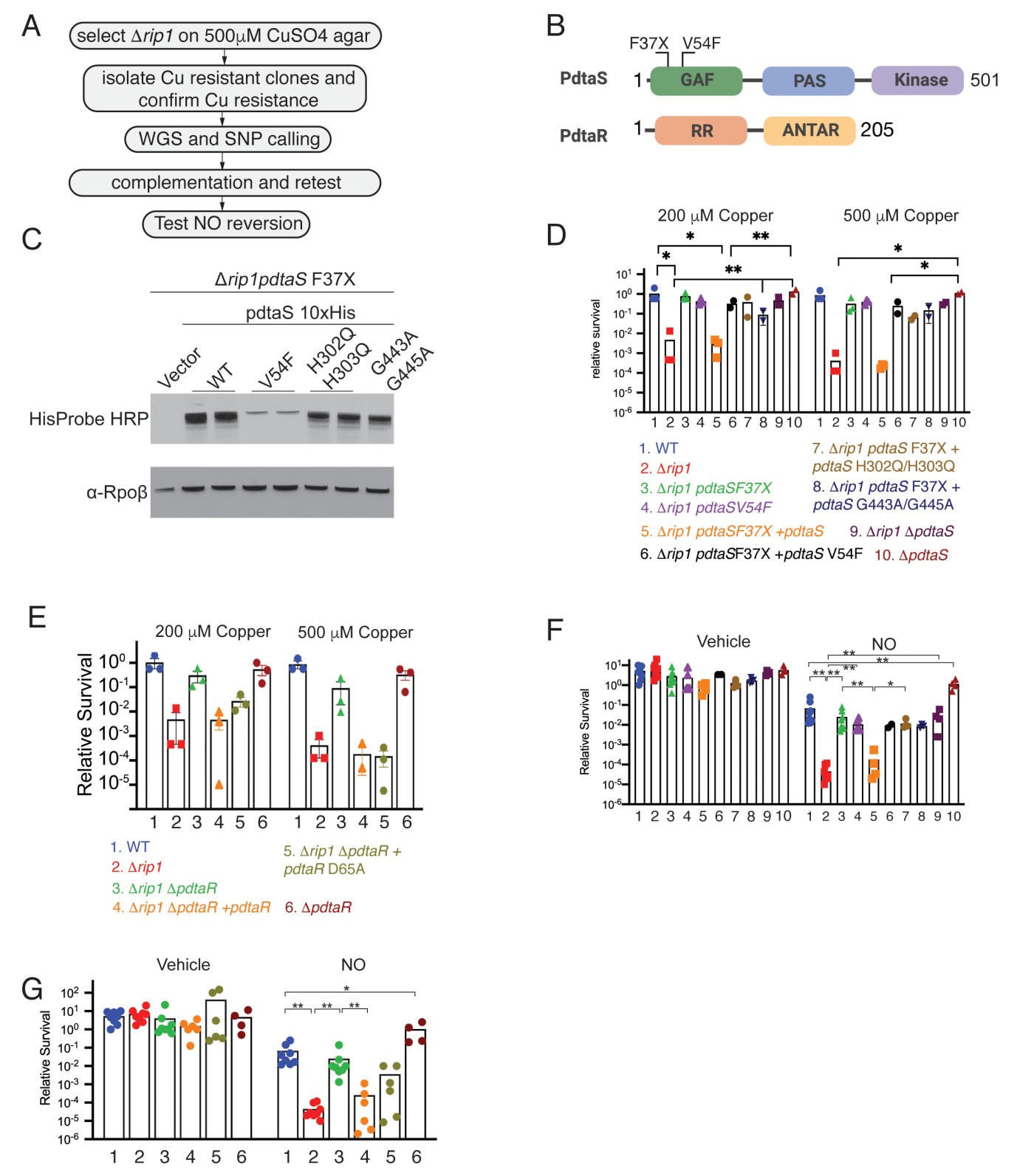

**Figure 2.** The PdtaS/PdtaR two-component system controls copper and NO resistance downstream of Rip1. (**A**) Flow chart of a genetic suppressor screen to isolate reversion mutations of the Rip1 Cu sensitivity and testing their reversion of NO sensitivity. WGS: whole genome sequencing; SNP: single nucleotide polymorphism. (**B**) Domain structure of PdtaS with GAF, PAS, and kinase domains. Identified suppressor mutations of PdtaS are shown in the N-terminal GAF domain. The reported phosphorylation target of PdtaS is the PdtaR response regulator (RR), which contains a C-terminal

*Figure 2 continued on next page*

Figure 2 continued

ANTAR RNA binding domain. (C) Protein levels of the indicated alleles of *pdtaS* reintroduced into the Δ*rip1pdtaS* F37X background as C-terminal 10X His fusion proteins with Rpoβ levels as a loading control. For WT, V54F, and H302Q/H303Q, two independent strains are shown, whereas for vector and G443A/G445A, one strain is shown. (D) Loss of *pdtaS* suppresses copper sensitivity of Δ*rip1*. Relative survival of the indicated strains grown on agar plates supplemented with 200 or 500 µM copper sulfate normalized as in *Figure 1*. H302Q/H303Q and G443A/G445A are kinase dead alleles. Each value is the average of technical duplicate measurements for n = 3 biological replicates. Statistical analysis by two-way ANOVA with Tukey's multi-comparison correction. (E) Loss of *pdtaR* suppresses copper sensitivity of Δ*rip1*. Cu sensitivity assay as noted in (D). PdtaR D65A lacks the receiver aspartate for PdtaS phosphorylation. Each value is the average of technical duplicate measurements for n = 3 biological replicates. (F) Loss of *pdtaS* suppresses NO sensitivity of Δ*rip1*. Colony-forming unit (CFU) counts of the indicated strains post treatment with vehicle or 200 µM diethylenetriamine nitric oxide (DETA-NO) for 3 days. Color coding of strain genotypes is the same as in (D). n = 8 biological replicates for WT, Δ*rip1*, Δ*rip1pdtaS* F37X; n = 6 biological replicates for Δ*rip1pdtaS* V54F; n = 4 biological replicates for Δ*rip1pdtaS* F37X + *pdtaS*, Δ*rip1pdtaS* F37X + *pdtaS* H302Q/H303Q, Δ*rip1*Δ*pdtaS*, Δ*pdtaS*; n = 3 biological replicates for Δ*rip1pdtaS* F37X + *pdtaS* G443A/G445A; n = 2 for Δ*rip1pdtaS* F37X + *pdtaS* V54F. Statistical analysis by Mann–Whitney test. (G) Loss of *pdtaR* suppresses NO sensitivity of Δ*rip1*. CFU counts of the indicated strains post treatment with vehicle or 200 µM DETA-NO for 3 days. Color coding of strain genotype is the same as in (E). n = 8 biological replicates for WT, Δ*rip1*, Δ*rip1* Δ*pdtaR*; n = 6 biological replicates for Δ*rip1*Δ*pdtaR* + *pdtaR* and Δ*rip1*Δ*pdtaR* +*pdtaR* D65A; n = 4 biological replicates for Δ*pdtaR*. Statistical analysis by Mann–Whitney test. For all panels, statistical significance is represented as *p<0.05, **p<0.01, ***p<0.001.

The online version of this article includes the following figure supplement(s) for figure 2:

**Figure supplement 1.** Chromosmoal PdtaS suppressor mutations and PdtaR levels under stress.

was recently reported to bind cyclic di-GMP (*Hariharan et al., 2021*), the full set of ligands that interact with the PdtaS sensing domains are not known. To determine whether PdtaS/R directly senses cytosolic metals and/or nitric oxide, we reconstituted the PdtaS/R phosphotransfer reaction using purified proteins. The PdtaS kinase was active for autophosphorylation when assayed with radiolabeled ATP (*Figure 3A*), as previously reported (*Morth et al., 2005*), and was also active in phosphotransfer to PdtaR (*Figure 3A*). PdtaS autophosphorylation was not inhibited by calcium or iron at 1 mM (*Figure 3—figure supplement 1A*), but we observed strong dose-dependent inhibition of PdtaS autophosphorylation at 10–1000 µM Cu (*Figure 3B*) or similar zinc concentrations (*Figure 3—figure supplement 1B,C*). Titrations revealed a dose-dependent inhibition of PdtaS activity by these metals, with inhibition constants (Ki) of 34 µM for Cu and 26 µM for zinc (*Figure 3B, F*, *Figure 3—figure supplement 1C*). These data indicate that copper and zinc directly inhibit the kinase activity of PdtaS.

We next asked whether NO has a similar effect on PdtaS. We used spermine NONOate, an NO donor with a half-life of NO release of 40 min. NO had a dose-dependent inhibitory effect on PdtaS activity, but spermine NONOate, which had been exhausted for NO release, had no effect (*Figure 3C*). The Ki determined from dose titrations was 7 µM. There was no effect of hydrogen sulfide on PdtaS activity (*Figure 3—figure supplement 1D*). Consistent with the idea that these ligands inhibit signaling without affecting expression or stability of the signaling proteins, we observed no change in PdtaR protein expression or proteolysis in vivo upon treatment with Cu, Zn, or NO (*Figure 2—figure supplement 1C*). These data indicate that PdtaS directly senses metals and NO and implies that ligand sensing by the sensor kinase inhibits signaling through the PdtaS/R system.

PdtaS contains GAF and PAS domains N-terminal to the kinase domain. PAS domains bind a wide variety of small molecule ligands (*Aravind and Ponting, 1997*; *Henry and Crosson, 2011*). As noted above, the two suppressor mutations in PdtaS are in the GAF domain, one a frameshift that inactivates the protein and one (V54F) in close proximity to a dicysteine motif (53-**C**VAQ**C**-57) (*Figure 2B*, *Figure 2—figure supplement 1*). To determine whether these domains are required for NO and Cu inhibition of PdtaS, we purified the PdtaS kinase domain, which was constitutively active (*Figure 3—figure supplement 1E,F*). However, neither Cu nor NO had a substantial inhibitory effect on the isolated kinase domain (*Figure 3—figure supplement 1E,F*), indicating that the GAF-PAS are required for the ligand-dependent inhibitory effect. We next mutated each cysteine in the PdtaS GAF in proximity to the V54F suppressor (C53A or C57A) and tested these kinases for Cu and NO inhibition. PdtaS-C53A was refractory to inhibition by both ligands (*Figure 3D–G*), directly implicating cysteine 53 in the GAF domain in Cu and NO sensing. PdtaS-C57A was also resistant to NO inhibition (*Figures 3G*, *Figure 3—figure supplement 2*), but our attempts to determine Cu inhibition of this protein were unsuccessful due to protein aggregation, preventing accurate quantitation (data not shown). These results indicate that the PdtaS sensor kinase is a constitutively active, ligand-

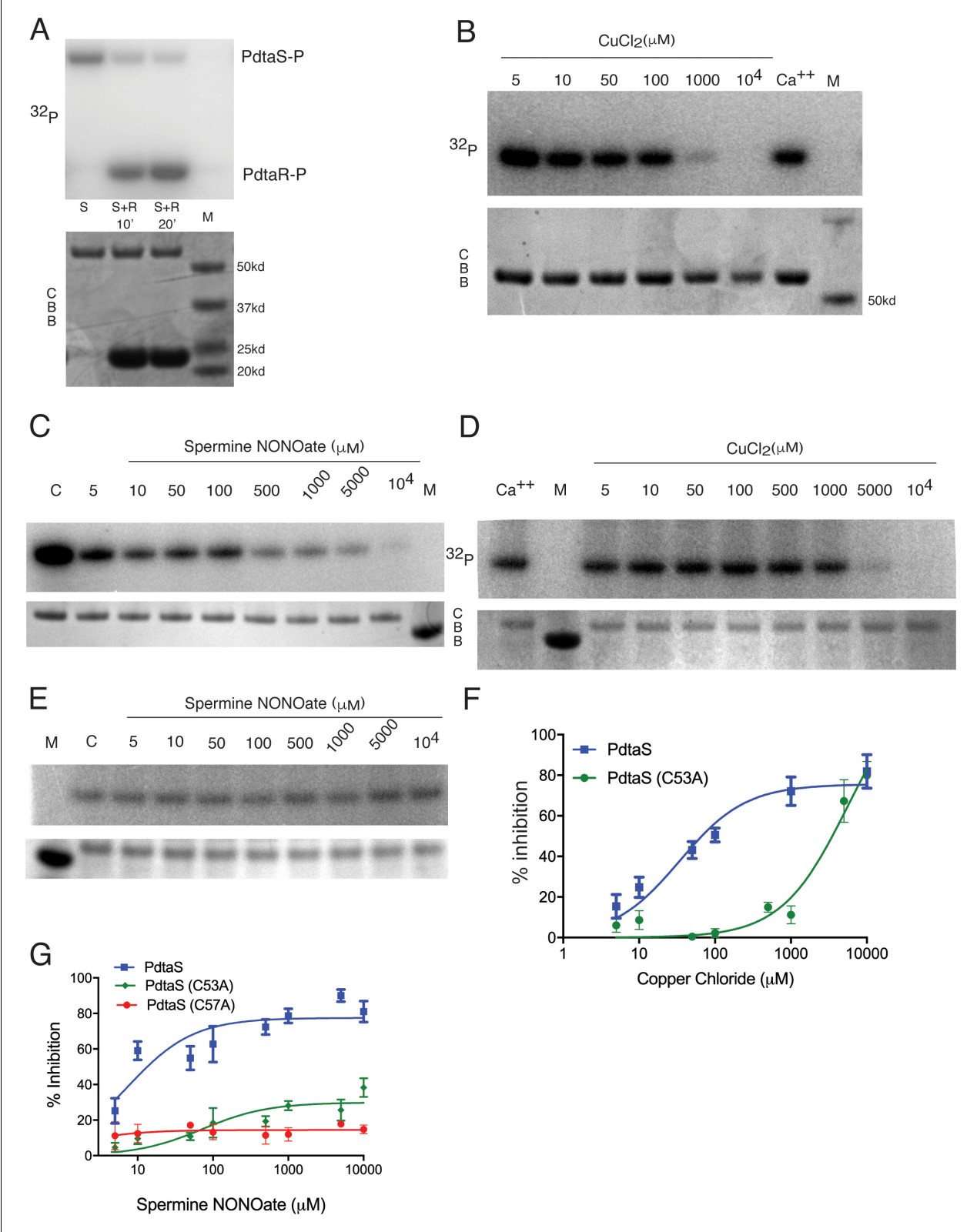

**Figure 3.** PdtaS is directly inhibited by copper and NO through a dicysteine motif in the N-terminal GAF domain. (**A**) PdtaS phosphotransfer to PdtaR. Upper panel: phosphorscreen imaging of $^{32}P$ incorporation; lower panel: Coomassie brilliant blue (CBB) staining to determine total protein. First reaction contains PdtaS alone, second and third lanes contain PdtaS/PdtaR incubated for 10 and 20 min, respectively. Fourth lane (M) is the molecular weight (MW) marker. (**B**) Cu$^{++}$ inhibits PdtaS autophosphorylation. Autophosphorylation of PdtaS protein preincubated with increasing

*Figure 3 continued on next page*

*Figure 3 continued*

concentrations (5 μM to $10^4$ μM) of $CuCl_2$ or 1 mM of $CaCl_2$ ($Ca^{++}$) as a control. Upper panel: phosphorscreen imaging of $^{32}P$ incorporation; lower panel: CBB staining to determine total protein. (C) NO inhibits PdtaS autophosphorylation. Autophosphorylation of PdtaS protein preincubated with increasing concentrations (5 μM to $10^4$ μM) of spermine NONOate and 1 mM of spermine NONOate post NO release as control (designated by C). (D) PdtaS GAF domain cysteine 53 is required for copper inhibition. PdtaS-C53A protein preincubated with increasing concentrations (5 μM to $10^4$ μM) of $CuCl_2$ or 1 mM of $CaCl_2$ was assayed for autophosphorylation as described in panel B . (E) PdtaS GAF domain cysteine 53 is required for NO inhibition. PdtaS-C53A protein preincubated with increasing concentrations (5 μM to $10^4$ μM) of spermine NONOate and of 1 mM of spermine NONOate post NO release as control was followed by autophosphorylation. (F) Cu inhibition curve from replicate data represented by (B) and (D) for PdtaS (blue) and PdtaS-C53A (green) proteins yields Ki of (34 ± 8 μM) for PdtaS and (5540 ± 2381 μM) for PdtaS-C53A. Error bars are SEM for n = 3. (G) NO inhibition curve from replicate data represented by (C) and (E) and (*Figure 3—figure supplement 2*) for PdtaS (blue), PdtaS-C53A (green), or PdtaS-C57A (red) proteins yields Ki of (7 ± 2 μM) for PdtaS and (73 ± 34 μM) for PdtaS-C53A. Error bars are SEM for n = 3 for WT and C53A and n = 2 for C57A.

The online version of this article includes the following figure supplement(s) for figure 3:

**Figure supplement 1.** PdtaS is inhibited by Zn and Cu through its GAF domain.
**Figure supplement 2.** PdtaS C57A is not inhibited by NO.

inhibited signaling protein that integrates Cu and NO sensing through a dicysteine motif in the GAF domain.

## Rip1/PdtaR jointly control an NO-responsive regulon that includes chalkophore biosynthesis

The data above indicates that PdtaS/R is a negative regulator of Cu and NO resistance and that these ligands directly inhibit PdtaS signaling. The exact function of PdtaR in regulating gene expression is not known. PdtaR-type RRs contain an ANTAR domain in place of the more commonly encountered DNA binding domain of traditional RR. In the few examples that have been examined, ANTAR domains bind to dual hairpin structures at the 5′ ends of mRNA, and thereby influence transcriptional termination (*Fox et al., 2009*), but the full spectrum of gene regulation conferred by ANTAR domain RNA binding has yet to be elucidated. ANTAR-RRs in Listeria and Enterococcus are also regulated by trans acting small RNAs that compete for ANTAR domain binding and thereby titrate the RR away from other mRNA targets (*DebRoy et al., 2014*; *Mellin et al., 2014*). To understand the target genes controlled by Rip1/PdtaS/PdtaR, we performed RNA sequencing under NO stress. We focused on NO stress for several reasons, including (1) the strong phenotypic reversion of Δ*rip1* in iNOS-deficient mice (*Figure 1G*), (2) the strong phenotypic reversion of Rip1 NO sensitivity by the *pdtaR* mutation, and (3) the technical advantages of treating Mtb with carefully titrated NO donors in liquid culture.

RNA sequencing of WT, Δ*rip1*, Δ*rip1*Δ*pdtaR*, and Δ*pdtaR* treated with vehicle or DETA-NO revealed that the NO-induced DosR regulon was intact in the Δ*rip1* strain and unaffected by loss of PdtaR (*Figure 4A*). However, by clustering gene expression across all strains and conditions, we defined a cluster of NO-induced, Rip1-dependent genes for which the defective expression is restored to WT levels in the Δ*rip1*Δ*pdtaR* strain, thereby matching the phenotypic pattern seen in our NO sensitivity tests (red box in *Figure 4B*). Consistent with these genes being negatively regulated by PdtaS/R in wild-type cells and the hyperresistance of the Δ*pdtaR* strain to NO (*Figure 2G*), this gene set was hyperinduced in the Δ*pdtaR* strain by NO (*Figure 4*). Importantly, there was no effect on basal gene expression with loss of *pdtaR*, indicating that inactivation of this TCS is not sufficient to activate gene expression in the absence of NO stress (*Figure 4B*).

Among the most strongly regulated genes controlled by Rip1/PdtaS/R under NO stress is a gene cluster from *Rv0096-0101*. This gene set encodes the PPE1 protein (*rv0096*) and five genes (*rv0097-0101*) that synthesize isonitrile lipopeptide chalkophores (*Harris et al., 2017*; *Wang et al., 2017*), which bind copper with high affinity (*Wang et al., 2017*; *Xu and Tan, 2019*). Alignment of RNA sequencing reads along the chalkophore cluster revealed low expression in basal conditions, but with a prominent small peak of 210 NT at the 5′ end of the *nrp* gene, indicating a site of potential termination (*Figure 4C*). With NO, chalkophore operon expression increased by three- to fivefold across the gene cluster in WT cells, but not in Δ*rip1* (*Figure 4C, D*). The defective chalkophore cluster expression is reversed in the Rip1/PdtaR double mutant. RNA reads mapping to the PPE1 locus, in contrast, mapped to a 336NT region that begins in the intergenic region at the reported transcription start site of the PPE1 mRNA and continues 277 nt beyond the PPE1 translational initiation

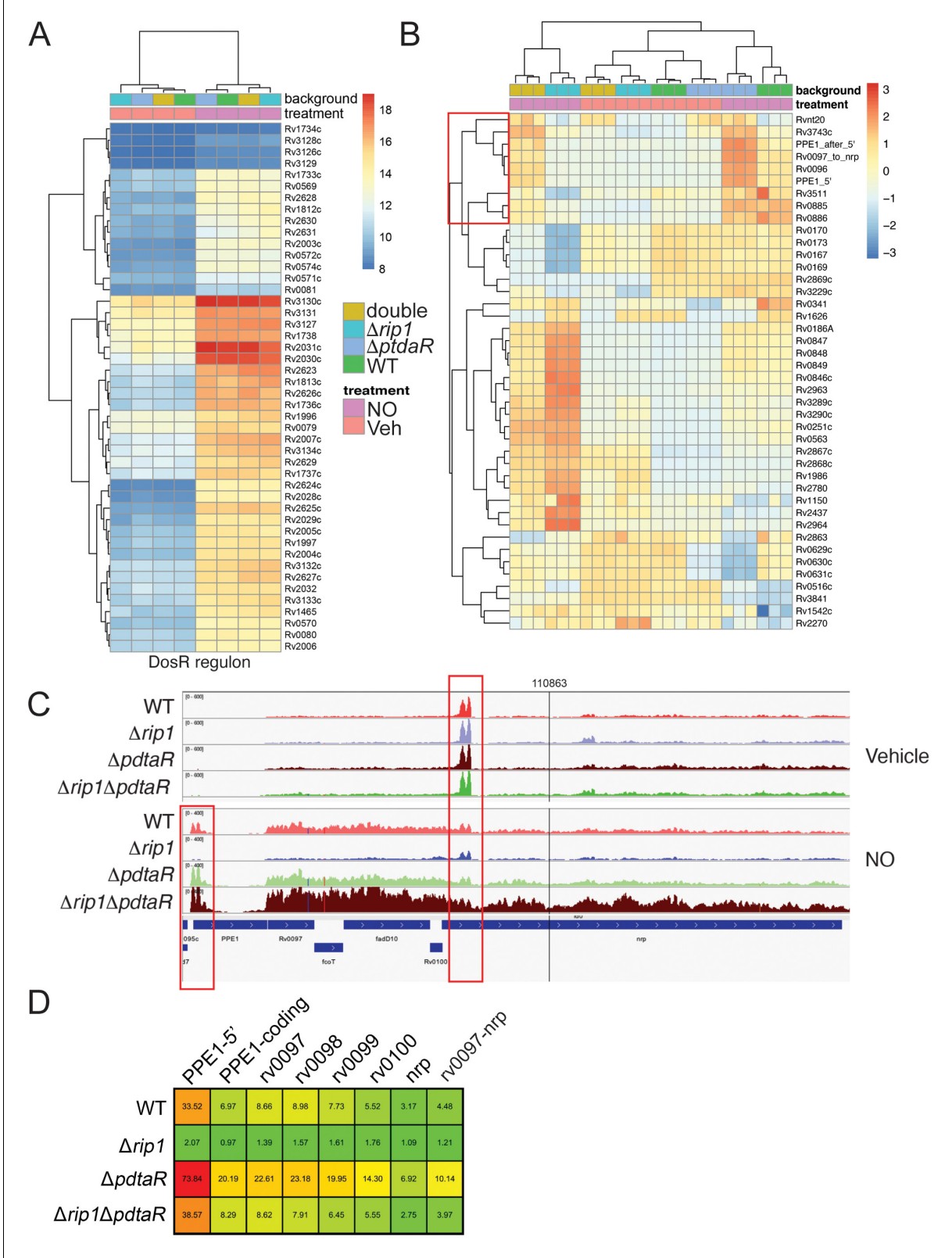

**Figure 4.** NO-induced dual regulation of the chalkophore biosynthetic operon by Rip1/PdtaS/PdtaR. (**A**) Unsupervised clustering of the diethylenetriamine nitric oxide (DETA-NO) (NO)-induced, DosR-regulated transcripts across the indicated genotypes. The scale bar represents the log₂ of the normalized counts for each gene. (**B**) Unsupervised clustering of gene expression of *M. tuberculosis* WT, Δ*rip1*, Δ*pdtaR*, and Δ*rip1ΔpdaR* using the same color coding as in (**A**) treated with vehicle (V) or DETA-NO (NO) The highlighted cluster indicated by the red box on the left includes PPE1-5′,
*Figure 4 continued on next page*

*Figure 4 continued*

the PPE1 coding sequences after PPE1 (PPE1-after 5′), and the *Rv0097-nrp* cluster as independent elements. The scale bar represents the log$_2$ of the scaled expression level for each row. The clustering of the genes and strains/conditions is done on the raw normalized counts. (C) Read coverage tracks across the *ppe1-nrp* cluster. Boxed areas highlight *ppe1-5′* peak at the 5′ end of PPE1 and a region of potential termination at 5′ end of *nrp*. (D) DETA-NO-induced fold change values (DETA-NO/vehicle) across the same coverage region as in (C). *0097-nrp* designates the entire *Rv0097-nrp* cluster.

codon (*Figure 4C*). We defined this element as 'PPE1-5′' and calculated its induction ratio with NO in comparison to the PPE1 coding sequence and the rest of the chalkophore operon. PPE1- 5′ is induced 33-fold by NO in wild-type cells, an induction that requires Rip1 (*Figure 4D*). PPE1-5′ is negatively regulated by PdtaR as evinced by hyperinduction in the Δ*pdtaR* strain. Loss of PdtaR restores the expression of PPE1-5′ in the Δ*rip1* background. Similar patterns were observed for each gene in the chalkophore operon and the chalkophore transcriptional unit as a whole (*Figure 4D*). These results identify a previously unappreciated NO-induced regulon that is subject to dual regulation by a positive transcriptional signal through the membrane-bound Rip1 protease and a negative signal from the cytoplasmic PdtaS/R TCS. Our data also identifies a previously unknown, NO-induced putative small RNA at the 5′ end of the chalkophore operon.

## PPE1-5′ binds directly to PdtaR

The data above indicate that Cu and NO resistance and PPE1-5′ expression in Mtb require a functional Rip1 and inhibition of the PdtaS/R system, which does not occur without Rip1. These cooperating systems jointly control chalkophore operon expression. One mechanism to unify these findings would be that the Rip1-controlled PPE1-5′ RNA binds directly to PdtaR to relieve the negative regulation of PdtaR on target genes such as *nrp*. As noted above, this mechanism of regulation by trans acting RNAs on ANTAR domain regulators has been reported previously (*DebRoy et al., 2014*; *Mellin et al., 2014*) and would explain why Rip1 is required to inactivate PdtaR activity. Inspection of the predicted RNA sequence of PPE-1-5′ for predicted RNA secondary structure that might bind PdtaR revealed three hairpins in the vicinity of the PPE-1 translational initiation codon (*Figure 5A*). To test whether this RNA may bind PdtaR, we produced a 284 nt PPE1-5′ RNA by in vitro transcription, along with positive (*Mehta et al., 2020*) and negative control RNAs (*Figure 5—figure supplement 1A,B*) and tested binding to PdtaR using microscale thermophoresis (MST). PdtaR was phosphorylated in vitro by PdtaS and binding was measured at varying concentrations of RNA. Negative control RNA derived from the *dnaK* gene did not bind PdtaR detectably (*Figure 5B*, *Figure 5—figure supplement 1C*) and the previously reported PdtaR binding RNA from *rv3864* (*Mehta et al., 2020*) bound with a K$_d$ of 2 μM (*Figure 5B*). We detected strong binding between phosphorylated PdtaR and PPE1-5′ with a calculated K$_d$ of 0.41 μM (*Figure 5B*). We tested the requirement for PdtaR phosphorylation in the PdtaR-PPE1-5′ interaction by omitting PdtaS. Surprisingly, given the usual requirement for phosphorylation in activating RR function, PdtaR binding to the rv3864 RNA was not affected by phosphorylation (*Figure 5C*). Unphosphorylated PdtaR still bound PPE1-5′, albeit with threefold lower affinity (K$_d$0.4 vs. 1.2 μM, *Figure 5C*), indicating that phosphorylation enhances binding but is not absolutely required.

## Chalkophore operon expression is the functional target that mediates Rip1/PdtaS/R-controlled NO resistance

To determine whether Rip1/PdtaS/PdtaR regulation of PPE1-5′ or the chalkophore cluster controls NO sensitivity, we restored expression of either the PPE1 coding sequence lacking the PPE1-5′ (hsp60-PPE1), PPE1 including PPE1-5′, or the chalkophore biosynthetic machinery (hsp60-*Rv0097-Rv0101*) in the Δ*rip1* strain. Restored expression of chalkophore biosynthesis (*rv0097-nrp*) restored NO resistance to a level equivalent to WT cells (*Figure 5D*). Enforced expression of PPE1 without PPE1-5′ had a minimal effect on NO sensitivity, whereas expression of PPE1 with PPE1-5′ restored NO resistance to the level similar to that observed with chalkophore re-expression (*Figure 5D*), despite no difference in the protein levels of PPE1 expressed from these two promoter constructs (*Figure 5—figure supplement 2A*). Despite the strong copper avidity of chalkophores, which might suggest a role in ameliorating Cu toxicity by direct binding, as is the case for other Cu binding molecules such as MymT, restoration of chalkophore production or expression of either PPE-5′ or PPE1 did not rescue the copper sensitivity of the Δ*rip1* strain (*Figure 5E*). Deletion of the *nrp* gene alone

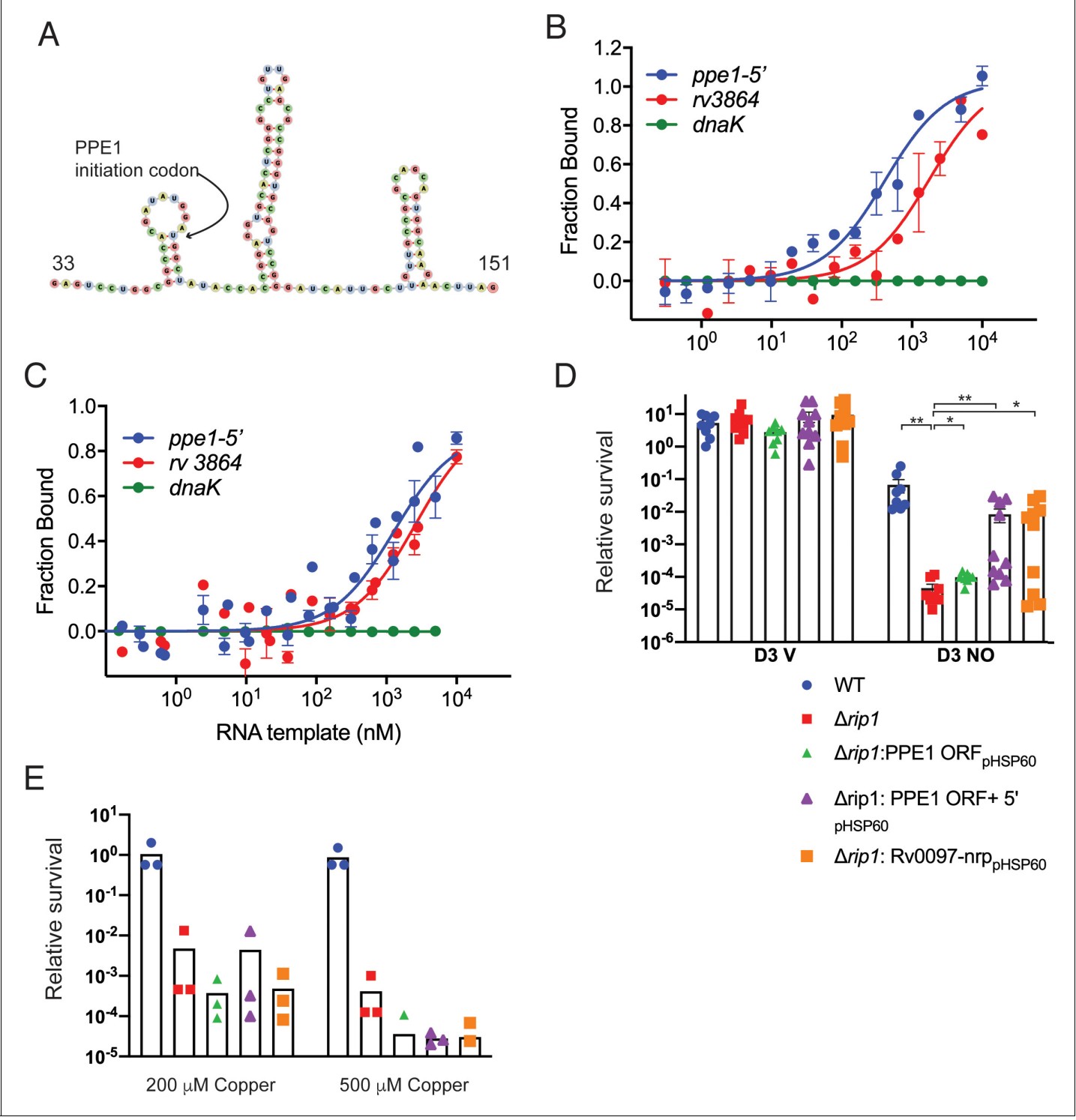

**Figure 5.** The PPE1-5′ RNA is an effector of NO resistance through sequestration of PdtaR. (**A**) Predicted structure of the RNA hairpins in PPE1-5′. The nucleotide numbering refers to distance from the first transcribed nucleotide of the *ppe1* RNA. The PPE1 translational initiation codon is indicated within the first hairpin. (**B**) Phosphorylated PdtaR directly binds to *ppe1*-5′. Change in the thermophoretic movement of fluorescently labeled and phosphorylated PdtaR was measured as a function of titrant RNA concentration on the X-axis for *ppe1*-5′, *rv3864,* and *dnaK* (ranging from 0.31 nM to 10 μM) as described in Materials and methods yielding binding affinity constants ($K_d$) for *ppe1-5′* = 410 ± 84 nM, *rv3864* = 2117 ± 798 nM while *dnaK* shows no binding. Error bars are SEM for n = 3. (**C**) RNA binding by PdtaR is partially phosphorylation dependent. Identical assay as in (**B**) using PdtaR without phosphorylation by PdtaS. Changes in the thermophoretic movement of fluorescently labeled PdtaR were measured as a function of titrant RNA concentration: *ppe1*-5′ (0.15 nM to 10 μM), *rv3864* (0.61 to 10 μM), and *dnaK* (0.15 nM to 10 μM). Binding affinities ($K_d$) are *ppe1-5′* = 1281 ± 322

*Figure 5 continued on next page*

*Figure 5 continued*

nM, *rv3864* = 2592 ± 832 nM while *dnaK* shows no binding. Error bars represent SEM for n = 3. (D) Rip1/PdtaS/PdtaR control of PPE1-5′ and chalkophore biosynthesis controls NO resistance. Constitutive expression of annotated protein coding region of *ppe1* alone (+*ppe1* ORF *hsp60*), *ppe1* coding region plus 222-nt 5′ of start codon (+*ppe1*-5′), both C-terminally fused to GFP, or the chalkophore cluster (+*rv0097-nrp*) in Δ*rip1* and testing for diethylenetriamine nitric oxide (DETA-NO) sensitivity. Relative survival of the indicated strains post treatment with vehicle or 200 µM DETA-NO. n = 8 biological replicates for WT, Δ*rip1*, Δ*rip1*: PPE1 ORF$_{pHSP60}$; n = 10 biological replicates for Δ*rip1*:PPE1 ORF +5′$_{pHSP60}$ and Δ*rip1*: Rv0097-nrp$_{pHSP60.}$ Statistical analysis by Mann–Whitney test. (E) Rip1/PdtaS/PdtaR control of PPE1-5′ and chalkophore biosynthesis does not control copper resistance. Relative survival of the same strains as in (D) grown on agar plates supplemented with 0, 200, or 500 µM copper sulfate normalized as in *Figure 1*. Each value is the average of technical duplicate measurements for n = 3 biological replicates. For all panels, statistical significance by t-test analysis is represented as *p<0.05, **p<0.01, ***p<0.001.

The online version of this article includes the following figure supplement(s) for figure 5:

**Figure supplement 1.** RNAs used in PdtaR binding experiments.

**Figure supplement 2.** PPE1-5′ UTR does not affect PPE1 protein expression and *nrp* loss is not sufficient for NO sensitivity.

did not alter NO resistance, indicating that there are multiple targets downstream of Rip1/PdtaS/R that contribute to NO resistance (*Figure 5—figure supplement 2B*). These results indicate that NO-induced expression of PPE1-5′ and chalkophore biosynthesis controlled by Rip1/PdtaS/PdtaR signaling are critical determinants of NO resistance in *M. tuberculosis*.

## Rip1/PdtaS/R control of chalkophore biosynthesis mediates acute lung infection

The model that emerges from the data presented above is that *M. tuberculosis* utilizes a two-signal mechanism to respond to the combinatorial stresses of the host: Rip1-dependent inactivation of PdtaR through the PPE1 5′ RNA and direct inhibition of PdtaS through the cysteine motif in the PdtaS GAF domain. This combined inhibition of PdtaS/R signaling relieves the negative regulation of this system on chalkophore biosynthesis. This model predicts that the in vivo virulence defect conferred by loss of Rip1, which is reversed by ablation of host NO (*Figure 1G*), should also be (1) phenocopied by loss of chalkophore biosynthesis (2) reversed in the Δ*rip1*/Δ*pdtaR* strain. We first tested the phenotype of *M. tuberculosis* lacking chalkophore biosynthesis (Δ*nrp*). Loss of *nrp* conferred mild attenuation in the early stages of lung growth such that Δ*nrp* titers were approximately 10-fold lower than wild type at 14 and 21 days after aerosol infection (*Figure 6A*). Loss of *nrp* also conferred more severe attenuation of extrapulmonary growth, which persisted during chronic infection (*Figure 6B*). We next infected mice by aerosol with Δ*rip1*/Δ*pdtaR* along with WT, Δ*rip1*, and Δ*pdtaR*. We observed that the severe attenuation of lung growth conferred by loss of Δ*rip1* was substantially reversed by loss of *pdtaR* such that bacterial lung titers of Δ*rip1*/Δ*pdtaR* were 15- and 97-fold higher than Δ*rip1* at 7 and 14 days post aerosol deposition, respectively (*Figure 6C*). Although lung titers of Δ*rip1*/Δ*pdtaR* were significantly lower than Δ*pdtaR*, which was attenuated compared to wild type, loss of *pdtaR* reversed the majority of the Rip1-dependent attenuation in the first two weeks of infection (*Figure 6C*). Importantly, the Rip1/PdtaR axis is only relevant to acute lung infection as loss of *pdtaR* did not reverse the severe attenuation of Δ*rip1* during chronic infection (*Figure 6C*) and had no effect on extrapulmonary tissues (*Figure 6D*), indicating that other pathways mediate these virulence phenotypes.

## Discussion

We have revealed a new branched pathway of signal transduction that integrates *M. tuberculosis* resistance to multiple host-derived stresses. *M. tuberculosis* is exposed to a mixture of toxic molecules within the host macrophage that include copper, zinc, and nitric oxide (*Botella et al., 2011*; *Darwin, 2015*; *Darwin et al., 2003*; *Darwin and Nathan, 2005*; *Shi et al., 2014*). Although signal transduction systems that respond to each of these molecules have been identified, the true in vivo determinants of *M. tuberculosis* resistance to these molecules are unknown. Additionally, although stresses are often assayed monolithically in vitro, this approach belies the situation in vivo, which the pathogen must sense and respond to simultaneous chemically distinct stresses that vary in intensity over time. Our studies identify an integrated sensing system that *M. tuberculosis* uses to directly sense combinations of stresses using a single sensing circuit.

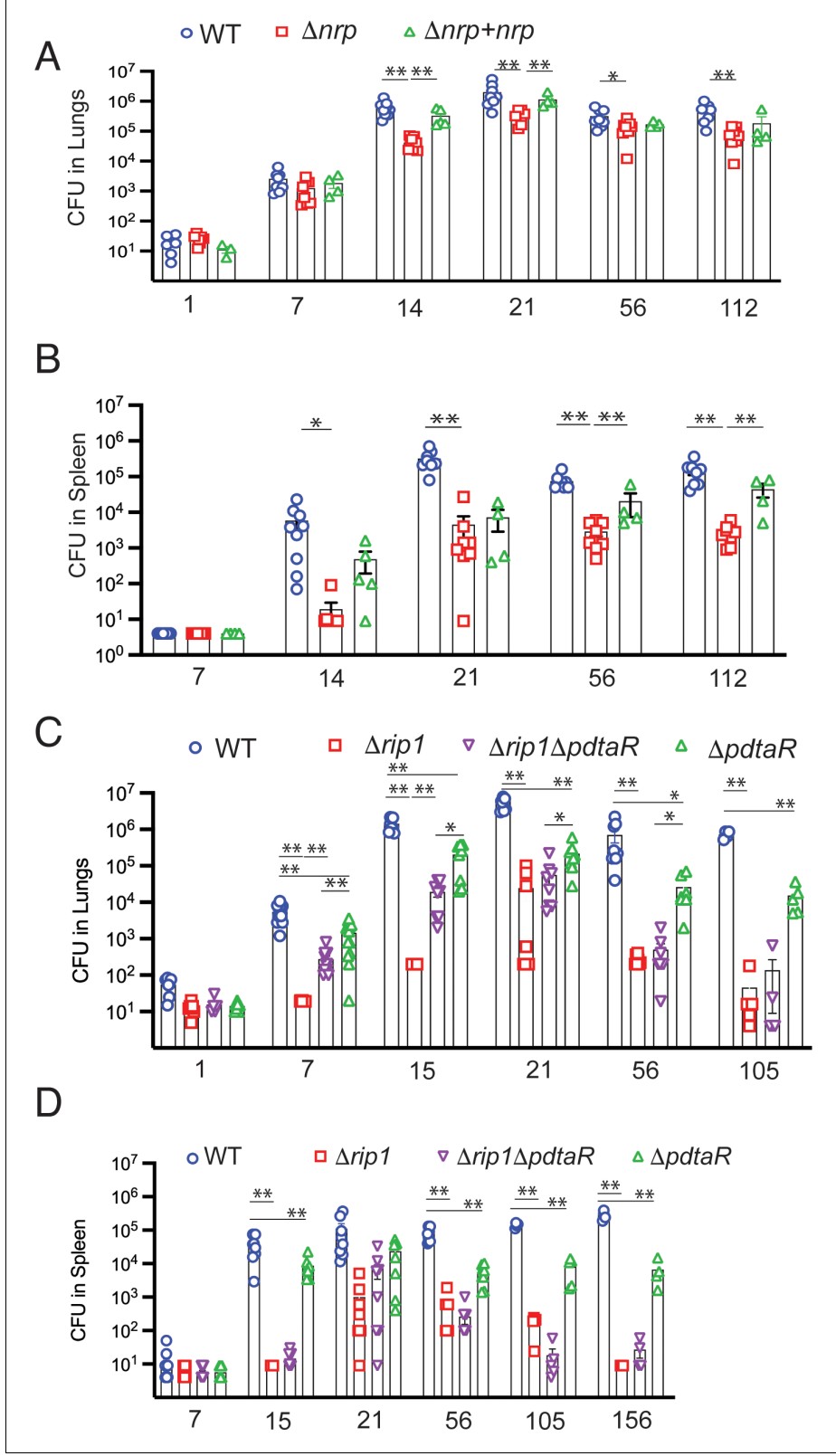

**Figure 6.** Rip1/PdtaS/PdtaR control of chalkophore biosynthesis controls early lung infection. (**A, B**) Bacterial burden (colony-forming unit [CFU]) of the indicated strains in lungs (**A**) or spleens (**B**) of C57bl/6 mice infected with WT (blue circle), Δ*nrp* (red square), or Δ*nrp +nrp* (green triangle) at the indicated time points (days) post aerosol infection. For WT and Δ*nrp*, n = 6 biological replicates on day 1 and n = 8 on all subsequent days. All p values
*Figure 6 continued on next page*

*Figure 6 continued*

calculated by unpaired t-test. (C, D) Bacterial burden (CFU) of the indicated strains in lungs (C) or spleens (D) of C57bl/6 mice infected with WT (blue circle), Δ*rip1* (red square), or Δ*rip1*Δ*pdtaR* (purple inverted triangle) or Δ*pdtaR* (green triangle) at the indicated time points (days) post aerosol infection. For all strains, n = 5 biological replicates on day 1; n = 12 biological replicates on day 7; n = 8 biological replicates on days 15, 21, and 56; n = 4 biological replicates on days 105 and 156. For all panels, statistical significance by unpaired Welch's t-test is represented as *p<0.05, **p<0.01.

---

The hub of this signaling system is the PdtaS/R TCS. This TCS is distinct from previously characterized TCS in *M. tuberculosis* in both structure and mechanism of regulation. The sensor kinase, PdtaS, is cytosolic rather than membrane bound. PdtaS is constitutively active in vitro (*Mehta et al., 2020*), and our data indicates the mechanism of signal transduction is ligand-induced inhibition of this constitutive activity. Rather than being activated by its ligands, PdtaS is inhibited by copper and NO through a dicysteine motif in the N-terminal GAF domain, defining an integrated mechanism by which two chemical effectors are sensed. The constitutively active PdtaS/R system negatively regulates a subset of genes required for NO resistance and relief of this inhibition by ligand-induced shutoff of PdtaS is necessary, but not sufficient, for expression of genes required NO resistance.

Our data also strongly demonstrates that full inhibition of PdtaS/R signaling requires the transduction of a cell surface signal by the intramembrane protease Rip1. Rip1 was originally identified as a S2P required for both acute growth in the mouse lung and persistence of the bacterium during chronic infection (*Makinoshima and Glickman, 2005*). Four anti-sigma factor substrates have been identified, but none of these previously identified pathways explain the virulence defects of *M. tuberculosis* lacking Rip1. The data presented here indicate that critical virulence function of Rip1 is to relieve PdtaR repression through control of a previously unrecognized small RNA at the 5′ end of the PPE1 gene. Rip1-controlled expression of PPE1 5′ is required for NO resistance as re-expression of this element is sufficient to reverse the NO sensitivity of Δ*rip1* and to re-express the chalkophore operon, itself sufficient to mediate NO resistance. The PPE1-5′ RNA binds directly to PdtaR in vitro, and this binding is enhanced, but does not absolutely require, PdtaR phosphorylation by PdtaS. This data, coupled with our genetic data, indicates that Rip1 control of PPE1-5′ directly inhibits PdtaR and de-represses PdtaR targets, which are ordinarily repressed by the PdtaS/R cascade. This mechanism of RNA-based titration of ANTAR domain RRs away from competitor target RNAs has been described as a mechanism of regulation in Enterococcus and Listeria (*DebRoy et al., 2014*; *Mellin et al., 2014*). This model suggests a branched logic of the Rip1/PdtaS/PdtaR system. Two signals are required to fully inactivate the PdtaS/R system and mediate NO resistance. A positive Rip1 transcriptional signal from the membrane to produce PPE1-5′ to sequester PdtaR, and a cytoplasmic inhibitory signal through the PdtaS kinase mediated by direct sensing of NO by the PdtaS GAF domain, which inhibits kinase activity. Our findings that PdtaR phosphorylation is not absolutely required for its function in NO resistance in vivo or RNA binding in vitro supports the model that two signals, PdtaS inhibition and PdtaR titration by the PPE1-5′ RNA, cooperate to titrate the activity of the PdtaS/R system. Our integrated model of this system is presented in *Figure 7*.

Our findings also clearly indicate that the coupled Rip1/PdtaS/R circuit controls Cu and NO resistance, but the gene sets downstream of this cascade that mediate resistance to each ligand are distinct. The NO resistance is due to regulation of isonitrile chalkophore biosynthesis. These lipopeptide molecules were recently defined in Streptomyces and *Mycobacterium marinum* as the products of the Nrp nonribosomal peptide synthase (*Harris et al., 2017*; *Wang et al., 2017*). The *nrp* gene has also been identified in multiple studies as a determinant of virulence (*Bhatt et al., 2018*), although the mechanism was not defined. The structure of the TB chalkophore is not known, but the Streptomyces molecule binds copper with extremely high affinity (*Xu and Tan, 2019*). Despite this Cu binding and an expectation that this property would predict a role in Cu resistance, our data instead indicate that these molecules are not acting to diminish copper toxicity and rather that their function is to ameliorate or prevent NO toxicity. The exact mechanisms by which chalkophores contribute to NO resistance in *M. tuberculosis* are not defined here, but in Methanotrophs chalkophores help maintain the activity of Cu-containing enzymes in the membrane (*Kenney and Rosenzweig, 2018*). An analogous function in mycobacteria might imply that NO stress displaces Cu from thiol metal centers, an activity that has been shown with MymT (*Gold et al., 2008*), and that

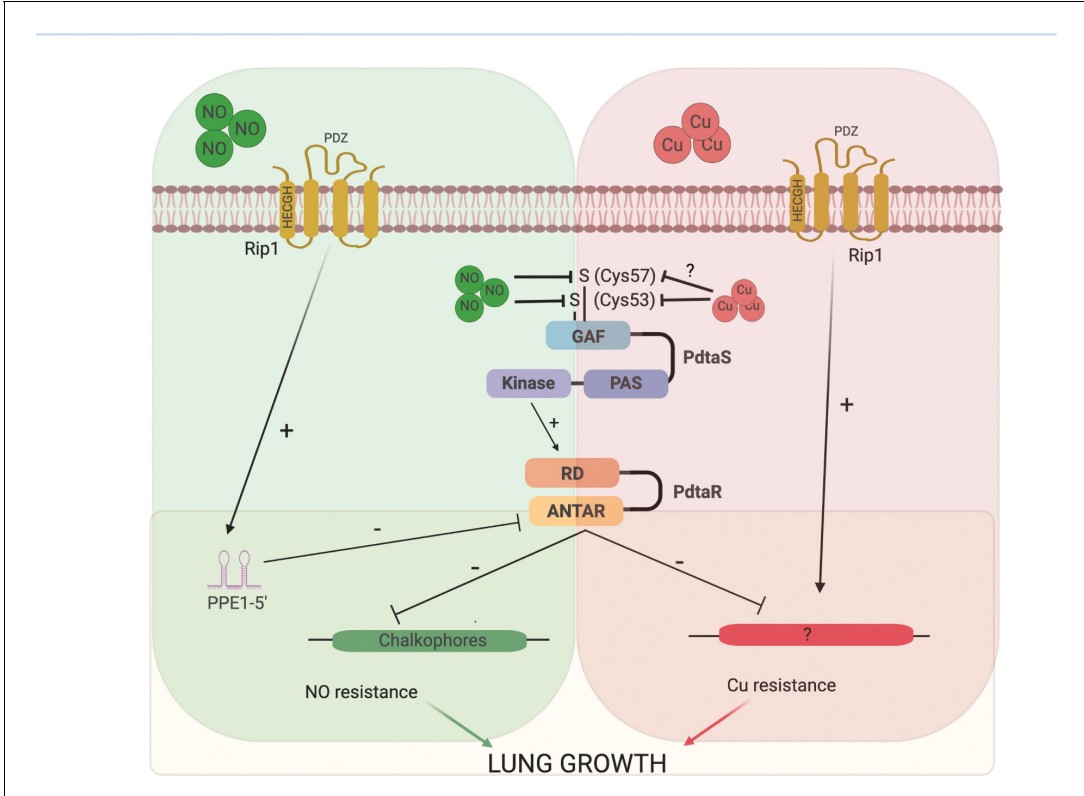

**Figure 7.** Model of the Rip1/PdtaS/PdtaR signaling circuit and its response to copper and nitric oxide. NO and copper resistance is mediated by cell surface and cytoplasmic convergent signals that inactivate PdtaS/PdtaR. In basal conditions, PdtaS/R is constitutively active, and that activity holds virulence genes, including chalkophore biosynthesis, inactive. NO or copper directly inhibit PdtaS activity through direct sensing by PdtaS N-terminal GAF domain. NO shutoff of PdtaS kinase activity requires C53 and 57, whereas Cu shutoff requires C53 with the role of C57 unknown. Sensing of NO at the cell surface contributes a second signal to relief of PdtaS/R by inducing Rip1-dependent expression of a hairpin containing small RNA at the 5′ end of the PPE1 gene (PPE1-5′), which binds directly to the ANTAR domain containing PdtaR. These signals relieve PdtaS/R repression of virulence gene expression, including chalkophore biosynthesis, to activate NO resistance. Although our data clearly implicates the upstream Rip1/PdtaS/PdtaR cascade in Cu resistance, the downstream targets that mediate Cu resistance are not known (indicated by ?). Despite the high Cu affinity of the isonitriles produced by the *nrp* locus, these molecules are involved in the NO resistance arm of the pathway rather than Cu. Figure constructed with BioRender.

chalkophores may compensate for this toxicity. This hypothesis and others will require further characterization of chalkophore functions in mycobacteria.

Although the Rip1/PdtaS/PdtaR cascade jointly controls NO and Cu resistance, we did not identify the chromosomal targets of PdtaS/R that mediate the Cu arm of the pathway. The Cu resistance controlled by this signaling cascade is distinct from known mechanisms of Cu binding or efflux based on our genetic data. Our data showing that NO-induced transcription of the PPE1-5′ RNA is the second signal in the NO resistance pathway, but not Cu resistance, leads us to hypothesize that as yet unidentified Rip1-controlled small RNA titrates away PdtaR from targets involved in Cu resistance. This model (*Figure 7*) implies that the target RNAs controlled by PdtaR may be modified based on the activating stress, a hypothesis that will be pursued in future work to determine the full complement of RNAs binding to PdtaR under Cu and NO stress.

In summary, our results identify a previously unrecognized mechanism of bacterial signal transduction that allows Mtb to rapidly adapt to the toxic environment of the host lung. This study opens a window into several important future questions, including the full spectrum of RNAs bound by the PdtaR RR, the ultimate mechanism by which this sensing circuit controls NO resistance, and the mechanisms of PdtaS sensing of ligand. These future questions will add additional detail to the critical mechanisms of TB pathogenesis identified here and potentially identify a critical pathway for therapeutic development against Mtb.

# Materials and methods

## Key resources table

| Reagent type (species) or resource | Designation | Source or reference | Identifiers | Additional information |
|---|---|---|---|---|
| Strain, strain background (*Mycobacterium tuberculosis*) | M.tb Erdman (WT, EG2) | Lab Stock | ATCC 35801 | Animal passaged |
| Strain, strain background (*Escherichia coli*) | DH5α | Lab Stock | ATCC SCC2197 | Plasmid maintenance strain |
| Strain, strain background (*Escherichia coli*) | EL350/ phAE87 | Lab Stock | | Phage packaging strain |
| Strain, strain background (*Escherichia coli*) | Rosetta 2 (DE3) | Millipore Sigma | Cat# 71397 | Recombinant protein expression strain |
| Strain, strain background (*Mus musculus*) female | C57BL/6J | Jackson Laboratory | Stock no: 000664; RRID:IMSR_JAX:000664 | |
| Strain, strain background (*Mus musculus*) female | B6;129P2-Nos2tm1Lau/J | Jackson Laboratory | Stock no: 002596; RRID:IMSR_JAX:002596 | |
| Gene (*M. tuberculosis*) | rip1 | ATCC 35801 | Erdman_3146 | |
| Gene (*M. tuberculosis*) | sigK | ATCC 35801 | Erdman_0488 | |
| Gene (*M. tuberculosis*) | sigL | ATCC 35801 | Erdman_0808 | |
| Gene (*M. tuberculosis*) | sigM | ATCC 35801 | Erdman_4291 | |
| Gene (*M. tuberculosis*) | sigD | ATCC 35801 | Erdman_3735 | |
| Gene (*M. tuberculosis*) | mymT | ATCC 35801 | Rv0186A | nt217703-217864 Erdman Genome |
| Gene (*M. tuberculosis*) | ctpV | ATCC 35801 | Erdman_1076 | |
| Gene (*M. tuberculosis*) | csoR | ATCC 35801 | Erdman_1074 | |
| Gene (*M. tuberculosis*) | nrp | ATCC 35801 | Erdman_0118 | |
| Gene (*M. tuberculosis*) | ppe1 | ATCC 35801 | Erdman_0113 | |
| Gene (*M. tuberculosis*) | rv0097 | ATCC 35801 | Erdman_0114 | |
| Gene (*M. tuberculosis*) | fcoT | ATCC 35801 | Erdman_0115 | |
| Gene (*M. tuberculosis*) | fadD10 | ATCC 35801 | Erdman_0116 | |
| Gene (*M. tuberculosis*) | rv0100 | ATCC 35801 | Erdman_0117 | |
| Gene (*M. tuberculosis*) | pdtaS | ATCC 35801 | Erdman_3533 | |

*Continued on next page*

*Continued*

| Reagent type (species) or resource | Designation | Source or reference | Identifiers | Additional information |
|---|---|---|---|---|
| Gene (*M. tuberculosis*) | *pdtaR* | ATCC 35801 | Erdman_1787 | |
| Genetic reagent (*M. tuberculosis*) | Δ*rip1* | *Sklar et al., 2010* | MGM3206; *rip1* KO | Chromosomal deletion of Erdman_3146 nt5-1212 by double crossover recombination followed by marker excision by LoxP recombination |
| Genetic reagent (*M. tuberculosis*) | Δ*sigD* | *Schneider et al., 2014* | JSS0005; *sigD* KO | Chromosomal deletion of Erdman_3735 nt6-622 by double crossover recombination |
| Genetic reagent (*M. tuberculosis*) | Δ*sigK* | *Sklar et al., 2010* | MGM3259; *sigK* KO | |
| Genetic reagent (*M. tuberculosis*) | Δ*sigL* | *Sklar et al., 2010* | MGM3254; *sigL* KO | |
| Genetic reagent (*M. tuberculosis*) | Δ*sigM* | *Sklar et al., 2010* | MGM3260, *sigM* KO | |
| Genetic reagent (*M. tuberculosis*) *sigL sigM* KO | Δ*sigL*Δ*sigM* | *Sklar et al., 2010* | MGM3255; *sigL sigM* KO | |
| Genetic reagent (*M. tuberculosis*) | Δ*sigK*Δ*sigL*Δ*sigM* | *Schneider et al., 2014* | MGM3256; *sigK sigL sigM* KO | |
| Genetic Reagent (*M. tuberculosis*) | Δ*sigK*Δ*sigL* | *Sklar et al., 2010* | MGM3261; *sigK sigL* KO | |
| Genetic reagent (*M. tuberculosis*) | Δ*sigK*Δ*sigM* | *Sklar et al., 2010* | MGM3283; *sigK sigM* KO | |
| Genetic reagent (*M. tuberculosis*) | Δ*sigK*Δ*sigD* | This study, available from corresponding author | *sigK*::*loxP sigD*::hygR; MGM3288 | Chromosomal deletion of Erdman_3735 nt6-622 by double crossover recombination MGM3259 background |
| Genetic reagent (*M. tuberculosis*) | Δ*sigL*Δ*sigD* | This study, available from corresponding author | *sigL*::*loxP sigD*::hygR; MGM3289 | Chromosomal deletion of Erdman_3735 nt6-622 by double crossover recombination MGM3254 background |
| Genetic reagent (*M. tuberculosis*) | Δ*sigM*Δ*sigD* | This study, available from corresponding author | *sigM*::loxP *sigD*::hygR; MGM3290 | Chromosomal deletion of Erdman_3735 nt6-622 by double crossover recombination MGM3260 background |
| Genetic reagent (*M. tuberculosis*) | Δ*ctpV* | This study, available from corresponding author | *ctpV*::hygR | Chromosomal deletion of nt 175-2136 of Erdman_1076 by double crossover recombination |
| Genetic reagent (*M. tuberculosis*) | Δ*csoR* | This study, available from corresponding author | *csoR*::hygR | Chromosomal deletion of nt 1-360 of Erdman_1074 by double crossover recombination |
| Genetic reagent (*M. tuberculosis*) | Δ*mymT* | This study, available from corresponding author | *mymT*::hygR | Chromosomal deletion of nt 1-162 of mymT by double crossover recombination |

*Continued on next page*

*Continued*

| Reagent type (species) or resource | Designation | Source or reference | Identifiers | Additional information |
|---|---|---|---|---|
| Genetic reagent (*M. tuberculosis*) | Δ*rip1*Δ*ctpV* | This study, available from corresponding author | *rip1::loxP ctpV::hygR* | Chromosomal deletion of nt 175-2136 of Erdman_1076 by double crossover recombination MGM3206 background |
| Genetic reagent (*M. tuberculosis*) | Δ*rip1*Δ*csoR* | This study, available from corresponding author | *rip1::loxP csoR::hygR* | Chromosomal deletion of nt 1-360 of Erdman_1074 by double crossover recombination MGM3206 background |
| Genetic reagent (*M. tuberculosis*) | Δ*rip1*Δ*mymT* | This study, available from corresponding author | *rip1::loxP mymT::hygR* | Chromosomal deletion of nt 1-162 of mymT by double crossover recombination MGM 3206 background |
| Genetic reagent (*M. tuberculosis*) | Δ*pdtaS* | This study, available from corresponding author | *pdtaS::hygR* | Chromosomal deletion of nt1-1506 of Erdman_3533 by double crossover recombination |
| Genetic reagent (*M. tuberculosis*) | Δ*pdtaS* + Vector | This study, available from corresponding author | *pdtaS::hygR: pMV306K* | Empty pMV306K integrated at AttB |
| Genetic reagent (*M. tuberculosis*) | Δ*pdtaR* | This study, available from corresponding author | *pdtaR::hygR* | Chromosomal deletion of nt 1-576 of Erdman_1787 by double crossover recombination |
| Genetic reagent (*M. tuberculosis*) | Δ*pdtaR* + vector | This study, available from corresponding author | *pdtaR::hygR:pMV306K* | Empty pMV306K integrated at AttB |
| Genetic reagent (*M. tuberculosis*) | Δ*nrp* | This study, available from corresponding author | *nrp::hygR* | Chromosomal deletion of nt 21-7515 of Edman_0118 by double crossover recombination |
| Genetic reagent (*M. tuberculosis*) | Δ*nrp* + vector | This study, available from corresponding author | *nrp::hygR: pMV306K* | Empty pMV306K integrated at AttB |
| Genetic reagent (*M. tuberculosis*) | Δ*nrp* + WT *nrp* | This study, available from corresponding author | *nrp::hygR: WT nrp* | WT copy of nrp integrated at AttB, Erdman_0116 promoter |
| Genetic reagent (*M. tuberculosis*) | Δ*rip1*Δ*pdtaS* | This study, available from corresponding author | *rip1::loxP pdtaS::hygR* | Chromosomal deletion of nt1-1506 of Erdman_3533 by double crossover recombination MGM3206 background |
| Genetic reagent (*M. tuberculosis*) | Δ*rip1*Δ*pdtaS* + vector | This study, available from corresponding author | *rip1::loxP pdtaS:: hygR: pMV306K* | Empty pMV306K integrated at AttB |
| Genetic reagent (*M. tuberculosis*) | Δ*rip1*Δ*pdtaR* | This study, available from corresponding author | *rip1::loxP pdtaR::hygR* | Chromosomal deletion of nt 1-576 of Erdman_1787 by double crossover recombination MGM3206 background |
| Genetic reagent (*M. tuberculosis*) | Δ*rip1*Δ*pdtaR* + vector | This study, available from corresponding author | *rip1::loxP pdtaR:: hygR: pMV306K* | Empty pMV306K integrated at AttB |
| Genetic reagent (*M. tuberculosis*) | Δ*rip1*Δ*pdtaR* + WT *pdtaR* | This study, available from corresponding author | *rip1::loxP pdtaR::hygR: WT pdtaR HA* | WT copy of pdtaR integrated at AttB |
| Genetic reagent (*M. tuberculosis*) | Δ*rip1*Δ*pdtaR* + D65A *pdtaR* | This study, available from corresponding author | *rip1::loxP pdtaR::hygR: D65A pdtaR HA* | D65A variant of pdtaR integrated at AttB |

*Continued on next page*

*Continued*

| Reagent type (species) or resource | Designation | Source or reference | Identifiers | Additional information |
|---|---|---|---|---|
| Genetic reagent (*M. tuberculosis*) | Δ*rip1*Δ*nrp* | This study, available from corresponding author | *rip1::loxP nrp::hygR* | Chromosomal deletion of nt 21-7515 of Edman_0118 by double crossover recombination MGM3206 background |
| Genetic reagent (*M. tuberculosis*) | WT + vector | *Makinoshima and Glickman, 2005* | EG2: pMV306K | Empty pMV306K integrated at AttB |
| Genetic reagent (*M. tuberculosis*) | WT + vector | *Makinoshima and Glickman, 2005* | EG2:pMV261 | Transformed with empty pMV261K episomal plasmid |
| Genetic reagent (*M. tuberculosis*) | Δ*rip1* + vector | *Sklar et al., 2010* | Δ*rip1*: pMV306K | Empty pMV306K integrated at AttB |
| Genetic reagent (*M. tuberculosis*) | Δ*rip1* + vector | *Sklar et al., 2010* | *rip1::loxP*: pMV261 | Transformed with empty pMV261K episomal plasmid |
| Genetic reagent (*M. tuberculosis*) | Δ*rip1* + WT Rip1 | *Sklar et al., 2010* *Makinoshima and Glickman, 2005* | *rip1::loxP*: WT *rip1* | Transformed with WT rip1 pMV261K episomal plasmid |
| Genetic reagent (*M. tuberculosis*) | Δ*rip1* + H21A Rip1 | *Makinoshima and Glickman, 2005* | *rip1::loxP*: H21A *rip1* | Transformed with H21A variant rip1 pMV261K episomal plasmid |
| Genetic reagent (*M. tuberculosis*) | Δ*rip1*: PPE1 ORF pHSP60 | This study, available from corresponding author | *rip1::loxP*: PPE1 GFP ORF only | Transformed with PPE1 GFP ORF only pMV261K episomal plasmid |
| Genetic reagent (*M. tuberculosis*) | Δ*rip1*: PPE1 ORF + 5' pHSP60 | This study, available from corresponding author | *rip1::loxP*: PPE1 GFP ORF + 5' UTR pHSP60 | Transformed with PPE1 GFP + 5' UTR pMV261K episomal plasmid |
| Genetic reagent (*M. tuberculosis*) | Δ*rip1*: *Rv0097-nrp* pHSP60 | This study, available from corresponding author | *rip1::loxP*: *Rv0097-nrp* | Transformed with Rv0097-nrp pMV261K episomal plasmid |
| Genetic reagent (*M. tuberculosis*) | Δ*rip1 pdtaS* F37X | This study, available from corresponding author | *rip1*::loxP *pdtaS* F37X | Isolated spontaneous Erdman_3533 variant in MGM3206 background |
| Genetic reagent (*M. tuberculosis*) | Δ*rip1 pdtaS* F37X + vector | This study, available from corresponding author | *rip1::loxP pdtaS* F37X: pMV306K | Empty pMV306K integrated at AttB |
| Genetic reagent (*M. tuberculosis*) | Δ*rip1 pdtaS* F37X + pdtaS | This study, available from corresponding author | *rip1::loxP pdtaS* F37X: WT *pdtaS* 10xHis | WT pdtaS 10xHis integrated at AttB |
| Genetic reagent (*M. tuberculosis*) | Δ*rip1 pdtaS* F37X + pdtaS V54F | This study, available from corresponding author | *rip1::loxP pdtaS* F37X: V54F *pdtaS* 10xHis | V54F variant of Erdman_3533 10xHis integrated at AttB |
| Genetic reagent (*M. tuberculosis*) | Δ*rip1 pdtaS* F37X + *pdtaS* H302Q/H303Q | This study, available from corresponding author | *rip1::loxP pdtaS* F37X: H302Q/H303Q *pdtaS* 10xHis | H302Q/H303Q variant of Erdman_3533 10xHis integrated at AttB |
| Genetic reagent (*M. tuberculosis*) | Δ*rip1 pdtaS* F37X + *pdtaS* G443A/G445A | This study, available from corresponding author | *rip1::loxP pdtaS* F37X: G443A/G445A *pdtaS* 10xHis | G443A/G445A variant of Erdman_3533 10xHis integrated at AttB |
| Genetic reagent (*M. tuberculosis*) | Δ*rip1 pdtaS* V54F | This study, available from corresponding author | *rip1::loxP pdtaS* V54F | Isolated spontaneous Erdman_3533 variant in MGM3206 background |
| Genetic reagent (*M. tuberculosis*) | Δ*rip1 pdtaS* V54F + vector | This study, available from corresponding author | *rip1::loxP pdtaS* V54F: pMV306K | Empty pMV306K integrated at AttB |

*Continued on next page*

*Continued*

| Reagent type (species) or resource | Designation | Source or reference | Identifiers | Additional information |
|---|---|---|---|---|
| Genetic reagent (*E. coli*) | Rosetta 2 DE3: pET WT PdtaS | This study, available from corresponding author | Rosetta 2 DE3: pET WT PdtaS | T7lac recombinant expression of C-terminally 10xHis tagged protein |
| Genetic reagent (*E. coli*) | Rosetta 2 DE3: pET WT PdtaS kinase domain | This study, available from corresponding author | Rosetta 2 DE3: pET WT PdtaS kinase domain | T7lac recombinant expression of C-terminally 10xHis tagged protein |
| Genetic reagent (*E. coli*) | Rosetta 2 DE3: pET C53A PdtaS | This study, available from corresponding author | Rosetta 2 DE3: pET C53A PdtaS | T7lac recombinant expression of C-terminally 10xHis tagged protein |
| Genetic reagent (*E. coli*) | Rosetta 2 DE3: pET C57A PdtaS | This study, available from corresponding author | Rosetta 2 DE3: pET C57A PdtaS | T7lac recombinant expression of C-terminally 10xHis tagged protein |
| Genetic reagent (*E. coli*) | Rosetta 2 DE3: pET WT PdtaR | This study, available from corresponding author | Rosetta 2 DE3: pET WT PdtaR | T7lac recombinant expression of C-terminally 10xHis tagged protein |
| Genetic reagent (*E. coli*) | Rosetta 2 DE3: pET D65A PdtaR | This study, available from corresponding author | Rosetta 2 DE3: pET D65A PdtaR | T7lac recombinant expression of C-terminally 10xHis tagged protein |
| Recombinant DNA reagent | pET SUMO (plasmid) | *Reverter and Lima, 2009* | pET SUMO | T7lac promoter *E. coli* expression vector N-terminal 10xHIS Sumo Fusion Protein |
| Recombinant DNA reagent | pET (plasmid) | This study, available from corresponding author | pET | T7lac promoter *E. coli* expression vector |
| Recombinant DNA reagent | PdtaS (plasmid) | This study, available from corresponding author | pET WT PdtaS 10XHIS | nt 1-1503 of Erdman_3533 fused N-terminal to Enterkinase cleavage site followed by 10xHis tag |
| Recombinant DNA reagent | PdtaS C53A (plasmid) | This study, available from corresponding author | pET C53A PdtaS 10XHIS | nt 1-1503 of Erdman_3533 C53A variant fused N-terminal to Enterkinase cleavage site followed by 10xHis tag |
| Recombinant DNA reagent | PdtaS C57A (plasmid) | This study, available from corresponding author | pET C57A PdtaS 10XHIS | nt 1-1503 of Erdman_3533 C57A variant fused N-terminal to Enterkinase cleavage site followed by 10xHis tag |
| Recombinant DNA reagent | PdtaS kinase domain (plasmid) | This study, available from corresponding author | pET WT PdtaS kinase domain 10XHIS | nt 678-1503 of Erdman_3533 variant fused N-terminal to Enterkinase cleavage site followed by 10xHis tag |
| Recombinant DNA reagent | PdtaR D65A (plasmid) | This study, available from corresponding author | pET Sumo WT PdtaR | N-terminal 10xHIS Smt3 fused to nt 1-615 of Erdman_1787 |
| Recombinant DNA reagent | PdtaR D65A (plasmid) | This study, available from corresponding author | pET Sumo D65A PdtaR | N-terminal 10xHIS Smt3 fused to nt 1-615 of Erdman_1787 D65A variant |
| Recombinant DNA reagent | Vector (plasmid) | *Sklar et al., 2010 Makinoshima and Glickman, 2005* | pMV261K | Episomal M.tb plasmid |
| Recombinant DNA reagent | Vector (plasmid) | *Sklar et al., 2010 Makinoshima and Glickman, 2005* | pMV306K | AttB integrating M.tb plasmid |
| Recombinant DNA reagent | pMV261K GFP (plasmid) | This study, available from corresponding author | pMV261K GFP | Episomal M.tb plasmid for C-terminal GFP fusion constructs |

*Continued on next page*

*Continued*

| Reagent type (species) or resource | Designation | Source or reference | Identifiers | Additional information |
|---|---|---|---|---|
| Recombinant DNA reagent | WT rip1 (plasmid) | *Makinoshima and Glickman, 2005* | pHMG121 | |
| Recombinant DNA reagent | H21A rip1 (plasmid) | *Makinoshima and Glickman, 2005* | pHMG141 | |
| Recombinant DNA reagent | pdtaS (plasmid) | This study, available from corresponding author | pMV306K WT PdtaS 10xHis | nt 1-1503 of Erdman_3533 tagged at C-term with 10x His |
| Recombinant DNA reagent | pdtaS V54F (plasmid) | This study, available from corresponding author | pMV306K V54F PdtaS 10xHis | nt 1-1503 of Erdman_3533 tagged at C-term with 10x His V54F variant |
| Recombinant DNA reagent | pdtaS H302Q/ H303Q (plasmid) | This study, available from corresponding author | pMV306K H302Q/H303Q PdtaS 10xHis | nt 1-1503 of Erdman_3533 tagged at C-term with 10x His H302Q/H303Q variant |
| Recombinant DNA reagent | pdtaS G443A/ G445A (plasmid) | This study, available from corresponding author | pMV306K G443A/H445A PdtaS 10xHIs | nt 1-1503 of Erdman_3533 tagged at C-term with 10x His G443A/G445A variant |
| Recombinant DNA reagent | pdtaR (plasmid) | This study, available from corresponding author | pMV306K WT PdtaR HA | nt 1-615 of Edman_1787 tagged at C-term with HA epitope |
| Recombinant DNA reagent | pdtaR D65A (plasmid) | This study, available from corresponding author | pMV306K D65A PdtaR HA | nt 1-615 of Edman_1787 tagged at C-term with HA epitope D65A variant |
| Recombinant DNA reagent | PPE1 ORF pHSP60 (plasmid) | This study, available from corresponding author | pMV261K ORF PPE1 GFP | nt 1-1389 of Erdman_0113 tagged at C-term with GFP |
| Recombinant DNA reagent | PPE1 ORF + 5' pHSP60 (plasmid) | This study, available from corresponding author | pMV261K ORF PPE1 GFP + 5'PPE1 | nt -222-1389 of Erdman_0113 tagged at C-term with GFP |
| Recombinant DNA reagent | Rv0097-nrp pHRP60 (plasmid) | This study, available from corresponding author | pMV261K Rv0097-nrp | nt 1 of Erdman_0114 to nt 7539 of Erdman_0118 |
| Recombinant DNA reagent | WT nrp (plasmid) | This study, available from corresponding author | pMV306 WT nrp | nt 1-96 of Erdman_0116 (predicted TSS Wadsworth) + nt 1-7539 of Erdman_0118 |
| Recombinant DNA reagent | pmsg360Hyg (plasmid) | *Sklar et al., 2010 Makinoshima and Glickman, 2005* | pmsg360Hyg | Phage packaging vector for hygR allelic exchange |
| Recombinant DNA reagent | pmsg360Zeo (plasmid) | *Sklar et al., 2010 Makinoshima and Glickman, 2005* | pmsg360Zeo | Phage packaging vector for zeoR allelic exchange |
| Recombinant DNA reagent | pmsg318-2 (plasmid) | *Sklar et al., 2010 Makinoshima and Glickman, 2005* | pmsg318-2 | M.tb Cre recombinase expression vector |
| Recombinant DNA reagent | pmsg360Hyg CtpV flanks (plasmid) | This study, available from corresponding author | Δ*ctpV* targeting vector | Hyg cassette flanked by 616 bp 5' CtpV nt 175 + 600 bp 3' CtpV nt 2136 |
| Recombinant DNA reagent | pmsg360 Hyg CsoR flanks (plasmid) | This study, available from corresponding author | Δ*csoR* targeting vector | Hyg cassette flanked by 600 bp 5' CsoR start codon + 600 bp 3' of CsoR stop codon |
| Recombinant DNA reagent | pmsg360 Hyg MymT flanks (plasmid) | This study, available from corresponding author | Δ*mymT* targeting vector | Hyg cassette flanked by 600 bp 5' MymT start codon + 600 bp 3' of MymT stop codon |

*Continued on next page*

*Continued*

| Reagent type (species) or resource | Designation | Source or reference | Identifiers | Additional information |
|---|---|---|---|---|
| Recombinant DNA reagent | pmsg360 Hyg *pdtaS* flanks (plasmid) | This study, available from corresponding author | Δ*pdtaS* targeting vector | Hyg cassette flanked by 600 bp 5' PdtaS start codon + 579 bp 3' PdtaS stop codon |
| Recombinant DNA reagent | pmsg360 Hyg pdtaR flanks (plasmid) | This study, available from corresponding author | Δ*pdtaR* targeting vector | Hyg cassette flanked by 600 bp 5' PdtaR start codon + 647 bp 3' PdtaR nt 576 |
| Recombinant DNA reagent | pmsg360 Hyg nrp Flanks (plasmid) | This study, available from corresponding author | Δnrp targeting vector | Hyg cassette flanked by 621 bp 5' nrp nt 21+ 621 bp 3' nrp nt 7515 |
| Sequence-based reagent | oSigA-1 | Integrated DNA Technologies | qPCR primer | cgtcttcatcccagacgaaat |
| Sequence-based reagent | oSigA-2 | Integrated DNA Technologies | qPCR primer | cgacgaagaccacgaagac |
| Sequence-based reagent | oCtpV-1 | Integrated DNA Technologies | qPCR primer | gtgtcccatgttcgaggtcaa |
| Sequence-based reagent | oCtpV-2 | Integrated DNA Technologies | qPCR primer | gtcaatgttcttcggtgcttac |
| Sequence-based reagent | oLpqS-1 | Integrated DNA Technologies | qPCR primer | gcatcgagttgtccaccag |
| Sequence-based reagent | oLpqS-2 | Integrated DNA Technologies | qPCR primer | tcaatgtggctcacccaaac |
| Sequence-based reagent | oMymT-1 | Integrated DNA Technologies | qPCR primer | gggtgatacgaatgacgaacta |
| Sequence-based reagent | oMymT-2 | Integrated DNA Technologies | qPCR primer | acagtggcatgggacttc |
| Sequence-based reagent | o2625c-F | Integrated DNA Technologies | qPCR primer | tcttgatcgcgttgggattg |
| Sequence-based reagent | o2625c-R | Integrated DNA Technologies | qPCR primer | cccggcgaattgatgtagag |
| Sequence-based reagent | oDesR-F | Integrated DNA Technologies | qPCR primer | tctgatcctcacgtcctacac |
| Sequence-based reagent | oDesR-R | Integrated DNA Technologies | qPCR primer | agcgcccacatctttgac |
| Sequence-based reagent | oHspX-F | Integrated DNA Technologies | qPCR primer | gaattcgcgtacggttccttc |
| Sequence-based reagent | oHspX-R | Integrated DNA Technologies | qPCR primer | gccaccgacacagtaagaatg |
| Sequence-based reagent | oPdtaS_C53A-F | Integrated DNA Technologies | PCR primer | gcgacgacggtgtcctggt ggcggttgcg |
| Sequence-based reagent | oPdtaS_C53A-R | Integrated DNA Technologies | PCR primer | cgcaaccgccaccaggac accgtcgtcgc |
| Sequence-based reagent | oPdtaS_C57A-F | Integrated DNA Technologies | PCR primer | gcgcaagcccggccgaac accgggccgacg |
| Sequence-based reagent | oPdtaS_C57A-R | Integrated DNA Technologies | PCR primer | cgtcggcccggtgttcggc cgggcttgcgc |
| Sequence-based reagent | oPdtaR_D65A-F | Integrated DNA Technologies | PCR primer | gtgatcatggccgtgaaga |
| Sequence-based reagent | oPdtaR_D65A-R | Integrated DNA Technologies | PCR primer | tcttcacggccatgatcac |

*Continued on next page*

*Continued*

| Reagent type (species) or resource | Designation | Source or reference | Identifiers | Additional information |
|---|---|---|---|---|
| Sequence-based reagent | oPdtaS_H302Q/H303Q-F | Integrated DNA Technologies | PCR primer | gggaaatccagcagcgggtt |
| Sequence-based reagent | oPdtaS_H302Q/H303Q-R | Integrated DNA Technologies | PCR primer | aacccgctgctggatttccc |
| Sequence-based reagent | oPdtaS_G443A/G445A-F | Integrated DNA Technologies | PCR primer | acgacgcgcttgctctgccg |
| Sequence-based reagent | oPdtaS_G443A/G445A-F | Integrated DNA Technologies | PCR primer | cggcagagcaagcgcgtcgt |
| Sequence-based reagent | Sense5'PPE1-F | Integrated DNA Technologies | PCR primer; in vitro transcription | taatacgactcactatataggg ggccgactaacaccgcgg |
| Sequence-based reagent | Sense5'PPE1-R | Integrated DNA Technologies | PCR primer; in vitro transcription | ggtttgctcaagccaggc |
| Sequence-based reagent | Rv3864-F | Integrated DNA Technologies | PCR primer; in vitro transcription | taatacgactcactataggcaaa aaattcgtgcaccaacc |
| Sequence-based reagent | Rv3864-R | Integrated DNA Technologies | PCR primer; in vitro transcription | tttccttacgctcgccgt |
| Sequence-based reagent | AntiDNAK-F | Integrated DNA Technologies | PCR primer; in vitro transcription | gatccacctagttctaga atggctcgtgcggtcg |
| Sequence-based reagent | AntiDNAK-R | Integrated DNA Technologies | PCR primer; in vitro transcription | taatacgactcactataggggg gcgtcattgaagtaggcg |
| Antibody | Anti *E. coli* RNA polymerase B antibody (α-Rpoβ) | BioLegend | 663903 (RRID:AB_2564414) | 1:10,000 dilution |
| Antibody | Anti-HA.11 epitope tag antibody (α-HA) | BioLegend | 901513 (RRID:AB_2565335) | 1:1000 dilution |
| Antibody | α-CarD (Rabbit) | Pocono Rabbit Farm | *Stallings et al., 2009* | 1:10,000 dilution |
| Antibody | α-MymT (Rabbit) | *Gold et al., 2008* | | 1:1000 dilution |
| Antibody | Anti-GFP(Rabbit) Antibody (α-GFP) | Rockland | 600-401-215L (RRID:AB_2612813) | 1:1000 dilution |
| Chemical compound, drug | Spermine NONOate | Millipore Sigma | 567703 | |
| Chemical compound, drug | Diethylenetriamine/nitric oxide adduct (DETA-NO) | Millipore Sigma | D185 | |
| Chemical compound, drug | Hydrogen peroxide, 30% | Fisher Scientific | H325-100 | |
| Chemical compound, drug | Sodium hydrosulfide hydrate | Fisher Scientific | AC296200250 | |
| Commercial assay, kit | In-Fusion HD Cloning Kit | Takara Bio USA | 639650 | |
| Commercial assay, kit | Maxima H Minus cDNA Synthesis Master Mix, with dsDNase | Thermo Fisher Scientific | M1681 | |
| Commercial assay, kit | DyNAmo Flash SYBR Green qPCR Kit | Thermo Fisher Scientific | F415F | |
| Commercial assay, kit | HiScribe T7 High Yield RNA Synthesis Kit | New England Biolabs | E2040S | |
| Commercial assay, kit | Ribo-Zero Magnetic Bacterial Kit | Epicentre | MRZB12424 | |
| Commercial assay, kit | TruSeq Stranded Total RNA kit | Illumina | 20020599 | |
| Commercial assay, kit | KAPA Hyper Prep Kit | Roche | 7962312001 | |

*Continued on next page*

*Continued*

| Reagent type (species) or resource | Designation | Source or reference | Identifiers | Additional information |
|---|---|---|---|---|
| Commercial assay, kit | TruSeq SBS Kit v3 | Illumina | FC-401-3002 | |
| Commercial assay, kit | GeneJET RNA Purification Kit | Fisher Scientific | FERK0731 | |
| Commercial assay, kit | TURBO DNA-free kit | Fisher Scientific | AM1907 | |
| Peptide, recombinant protein | Phusion High Fidelity Polymerase | Fisher Scientific | F530L | |
| Commercial assay, kit | GeneJET Plasmid Miniprep Kit | Fisher Scientific | FERK0503 | |
| Commercial assay, kit | NanoTemper Technologies Inc PROTEIN LABELING KIT RED-NHS (MOL011) | Fisher Scientific | NC1491187 | |
| Commercial assay, kit | NanoTemper Technologies Inc Standard capillaries | Fisher Scientific | NC1408770 | |
| Software, algorithm | ViennaRNA Web Services | | http://www.viennarna.at/forna/; RRID:SCR_008550 | |
| Software, algorithm | fastqc | http://www.bioinformatics.babraham.ac.uk/projects/fastqc | RRID:SCR_014583 | |
| Software, algorithm | bwa mem | *Li et al., 2009* | RRID:SCR_010910 | |
| Software, algorithm | samtools | *Li et al., 2009* | RRID:SCR_002105 | |
| Software, algorithm | Bioconductor Rsubread package | *Liao et al., 2019* | RRID:SCR_016945 | |
| Software, algorithm | DESeq2 R package | *Love et al., 2014* | RRID:SCR_015687 | |
| Software, algorithm | R | The R Project for Statistical Computing; https://www.R-project.org/ | RRID:SCR_001905 | |
| Software, algorithm | GATK | Broad Institute; https://gatk.broadinstitute.org/hc/en-us | RRID:SCR_001876 | |
| Software, algorithm | pheatmap | https://www.rdocumentation.org/packages/pheatmap/versions/0.2/topics/pheatmap | RRID:SCR_016418 | |
| Other | NUPAGE 4-12% BT Gel | Fisher Scientific | NPO312BOX/ NPO336BOX | |
| Other | Protran Nitrocellulose Hybridization Transfer Membrane | Perkin Elmer | NBA08C001EA | |
| Other | Immun-Blot PVDF Membrane | Bio-Rad | 1620177 | |
| Other | NanoTemper Technologies Premium Capillaries | Fisher Scientific | NC1408772 | |
| Other | HisProbe-HRP Conjugate | Thermo Fisher Scientific | 15165 | |

## Reagents

Media and salts were purchased from Thermo Fisher, USA. Glycine, imidazole, and ATP from Merck Sigma-Aldrich, USA. Protein marker from Thermo Scientific, USA. Antibiotics, isopropyl β-D-1-thiogalactopyranoside (IPTG) and dithiothreitol (DTT) from GoldBio Inc, USA. Protease inhibitor cocktail from Amresco, USA. Ni$^{2+}$-NTA resin from Qiagen, GmBH. $\gamma^{32}$P-labeled ATP (>3000 Ci/mmol) from Perkin Elmer, USA.

## General growth conditions, strains, and DNA manipulations

*M. tuberculosis* (Erdman) and derivatives were grown and maintained in 7H9 media (broth) or on 7H10 (agar) supplemented with 10% Oleic Acid-Albumin-Dextrose-Catalase supplement (OADC), 05% glycerol, and 0.05% Tween-80(broth only) (7H9 OADC/7H10 OADC) at 37°C unless otherwise noted below. The wild-type strain used in this study is animal passaged, minimally passaged in vitro, and confirmed to be phthicerol dimycocerosates (PDIM) positive both by metabolic labeling and without mutations in PDIM biosynthesis by whole genome sequencing. Deletion mutations were generated by specialized transduction utilizing the temperature-sensitive phage phAE87. Mutant strains were confirmed by Southern blot or PCR using primers in the Hyg cassette and outside the cloned region, followed by sequencing of the amplified PCR product to confirm the location of chromosomal insertion. For a complete strain list with relevant features, see *Supplementary file 1*. Plasmids utilized in this study were generated using standard molecular techniques and are listed with their features in *Supplementary file 2*.

## Metal sensitivity assays

For agar-based assays, duplicate logarithmically growing (OD$_{600}$ of 0.4–0.6) cultures were centrifuged and washed twice with room temperature phosphate buffered saline (PBS) containing 0.05% Tween-80 (PBS Tween). Resulting washed suspensions were all adjusted to OD$_{600}$ of 0.2 and serial diluted in PBS Tween. Dilutions were then plated onto control unsupplemented 7H10 OADC agar media, as well as media supplemented with the indicated concentration of copper (II) sulfate or iron (III) chloride. Colony-forming units (CFUs) were enumerated after 28 days of growth at 37°C, 5% CO$_2$. Relative survival was calculated by dividing the average number of CFU on treatment plates by the average number of CFU on control plates. Experiments were repeated a minimum of two biological replicates defined as individual samples treated independently throughout. Where applicable, technical replicate refers to repeated quantification of identical biological samples to account for error during dilutions for agar plate cultures.

Liquid growth curve assays were pre-grown and washed as above. Growth curves were started at an initial OD$_{600}$ of 0.005 by adding 1 mL of washed culture, at an OD$_{600}$ of 0.05, to 9 mL of 7H9 OADC supplemented with 100 µM zinc (II) sulfate. Biological replicates defined as individual samples treated independently throughout.

For copper liquid sensitivity CFU assays, 25 mL cultures were pre-grown to an OD$_{600}$ of 0.8–1.0 in 7H9 OADC media. Cultures were washed twice in 7H9 supplemented with albumin dextrose saline, 05% glycerol, 02% Tween-80 (7H9 ADS), and normalized to an OD$_{600}$ of 0.1 in four-replicate 10 mL cultures. After equilibration at 37°C for 2 hr, duplicate cultures were treated with 200 µM copper sulfate (Sigma-Aldrich), while control cultures were left untreated. On days 0 and 3 post treatment, aliquots of each culture were removed, washed, and serial diluted in PBS Tween. Dilutions were then plated on 7H10 OADC plates for CFU determination. CFUs were enumerated after 28 days of growth at 37°C, 5% CO$_2$. Relative survival was calculated by dividing the average CFU day 3 by the average number of CFU on day 0. Biological replicates defined as individual samples treated independently throughout.

## Nitric oxide sensitivity assays

Duplicate 25 mL cultures were pre-grown to an OD$_{600}$ between 0.8 and 1.0 in 7H9 OADC media. Cultures were washed twice in 7H9 ADS and normalized to an OD$_{600}$ of 1.0 in 10 mL of 7H9 ADS. Four-replicate, 9 mL 7H9 ADS cultures were then inoculated with 1 mL of this suspension for a starting OD$_{600}$ of 0.1 in 30 mL inkwell bottles (Fisher Scientific Cat# 03-313-89A) with ~2 in. of headspace. After equilibration at 37°C for 2 hr, duplicate cultures were treated with 100 µL of freshly prepared 20 mM DETA-NO, or 0.1 M sodium hydroxide (final concentration 1mM, vehicle for DETA-

NO). Treatment was repeated every 24 hr for 3 days. Days 0 and 3 post treatment, aliquots of each culture were removed, washed, and serial diluted in PBS Tween. Dilutions were then plated on 7H10 OADC plates for CFU determination. CFUs were enumerated after 28 days of growth at 37°C, 5% $CO_2$. Relative survival was calculated by dividing the average CFU day 3 by the average number of CFU on day 0. Experiments were repeated a minimum of two times. Biological replicates defined as individual samples treated independently throughout.

### PBS starvation
Cultures of the indicated strains were pre-grown in 7H9 OADC media to an OD of between 0.5 and 0.7. Cells were then collected by centrifugation and washed twice with PBS pH 7.0. Replicate flasks containing 30 mL of PBS were then inoculated to an OD of 0.1. Flasks were incubated standing in an incubator at 37°C with 5% $CO_2$. At the indicated times, aliquots were removed, serially diluted in PBS, and plated onto 7H10 OADC agar plates. CFUs were enumerated after 28 days of incubation at 37°C. Biological replicates defined as individual samples treated independently throughout.

### Lysozyme, hydrogen peroxide, and SDS sensitivity
Cultures of the indicated strains were pre-grown in 7H9 OADC to an OD of between 0.5 and 0.7. Cells were then washed twice with 7H9 ADS media and diluted to an OD of 0.1 in the same media. 1 mL of washed culture was then inoculated into replicate 9 mL bottles of fresh 7H9 ADS media (untreated control) or 7H9 ADS media containing 2.5 mg/mL lysozyme, 10 mM hydrogen peroxide, or 0.05% (v/v) sodium dodecyl sulfate (SDS). Following 3 hr of treatment at 37°C, aliquots from each culture were removed, serial diluted in 7H9 ADS media, and then spread onto 7H910 OADC plates. CFUs were enumerated following 28 days of incubation at 37°C. Biological replicates defined as individual samples treated independently throughout.

### Survival at pH 4.5
Cultures of the indicated strains were pre-grown in 7H9 OADC to an OD of between 0.5 and 0.7. Cultures were then divided in half and washed with either 7H9 ADS-pH 7.0 or 7H9 ADS-pH 4.5. Replicate 10 mL bottles of pH 4.5 or pH 7.0 media were then inoculated with washed culture at on OD of 0.01. Aliquots of each culture were removed immediately after inoculation, and after 7 days of growth at 37°C, serial diluted, and then plated onto 7H10 OADC plates. CFUs were enumerated following 28 days of incubation at 37°C. Biological replicates defined as individual samples treated independently throughout.

### Copper-induced MymT expression
MymT expression was induced as follows: cultures grown in 7H9 OADC to an $OD_{600}$ of ~0.5 were divided into 20 mL replicate aliquots. Copper sulfate (0–500 µM) was then added to individual replicates followed by 2 hr incubation at 37°C. Following incubation, cells were processed for immunoblot as described in Materials and methods, excepting that the PVDF membrane was used for blotting.

RNA extraction and RT-qPCR mRNA levels were quantified as follows. For copper-stimulated transcripts, triplicate 40 mL cultures grown to an $OD_{600}$ of 0.4 in 7H9 OADC were divided into two sets of 3, 20 mL cultures. Half were left untreated for quantification of basal transcript level, while the other half were treated with 200 µM copper (II) sulfate for 2 hr.

For nitric oxide-stimulated transcripts, triplicate 40 mL culture grown to an $OD_{600}$ of 0.4 in 7H9 ADS were divided into two sets of 3X20 mL cultures. Half were then treated with 200 µL of 20 mM DETA-NO dissolved in 0.1 M sodium hydroxide for 3 hr or with vehicle (0.1 M sodium hydroxide)-alone control. Cells were collected by centrifugation, washed once in PBS Tween, and then suspended in 1 mL of Trizol (Invitrogen). Cells were mechanically disrupted with zirconia bead via 3 × 30 s pulses in a BioSpec Mini24 beadbeater. Total RNA was then extracted according to the manufacturer's instructions. Contaminating genomic DNA was removed using the Turbo DNA-free kit (Invitrogen), and RNA was then further purified utilizing RNeasy Mini spin columns (Qiagen). 1 µg of resulting total RNA was used to synthesize cDNA via random priming utilizing the Maxima H-Minus cDNA Synthesis kit (Thermo Fisher). Real-time qPCR was performed on a 7500 real-time PCR system (Applied Biosystems). Amplification product was detected by SYBR green using the Dynamo Flash

qPCR kit (Thermo Fisher). For each gene of interest (GOI)-normalized cycle threshold, C(t) was determined relative to the housekeeping gene *sigA*. Relative expression level was calculated using the formula $2^-(C(t)GOI-C(t)sigA)$. Primer sets used to amplify individual GOI are listed in *Supplementary file 3*.

## Aerosol infection of mice

Mice used in this study were purchased from The Jackson Laboratory. 8–10-week-old C57Bl/6J (stock number 00064) were used. iNOS ko mice were purchased from Jackson, stock number 002609. All purchased mice were rested within our animal facility to normalize microbiota for 2 weeks. Care, housing, and experimentation on laboratory mice were performed in accordance with the National Institute of Heath guidelines and the approval of the Memorial Sloan-Kettering Institutional Animal Care and Use Committee (IACUC). Protocol approval number is 01-11-030.

Strains for infection were grown to an $OD_{600}$ of 0.5–0.7 in 7H9 OADC. They were then washed twice with PBS Tween followed by brief sonication to disrupt aggregates. Final inoculums were prepared by suspending $8 \times 10^7$ CFU in 10 mL of sterile water. Appropriate mouse strains were then exposed to $4 \times 10^7$ CFU in a Glas-Col aerosol exposure unit. At the indicated time point post infection, 4–8 individual mice were humanely euthanized and both lungs and spleens were harvested for CFU determination. Organ homogenates were cultured on 7H10 OADC plates, and CFU was enumerated after 28 days of incubation at 37°C, 5% $CO_2$.

## Suppressor screen and whole genome sequencing

Spontaneous copper suppressor mutants were identified by selecting ~$3 \times 10^5$ CFU of Δ*rip1* bacteria on each 7H10 OADC agar dish supplemented with 500 µM copper sulfate. A total of approximately $5x10^7$ bacteria were selected. Plates were then incubated at 37°C, 5% $CO_2$ for 28 days. Individual genetic suppressor candidates were picked and expanded to an $OD_{600}$ of 0.5 in 40 mL of 7H9 OADC without addition of copper followed by confirmation of Cu resistance. Whole genome sequencing was performed acoustically shearing genomic DNA, and HiSeq sequencing libraries were prepared using the KAPA Hyper Prep Kit (Roche). PCR amplification of the libraries was carried out for 10 cycles. $5-10 \times 10^6$ 50 bp paired-end reads were obtained for each sample on an Illumina HiSeq 2500 using the TruSeq SBS Kit v3 (Illumina). Post-run demultiplexing and adapter removal were performed and fastq files were inspected using fastqc (Andrews S. (2010). FastQC: a quality control tool for high-throughput sequence data; available at: http://www.bioinformatics.babraham. ac.uk/projects/fastqc). Trimmed fastq files were then aligned to the reference genome (*M. tuberculosis* str. Erdman; GenBank: AP012340.1) using bwa mem (*Li and Durbin, 2009*). Bam files were sorted and merged using Samtools (*Li et al., 2009*). Read groups were added and bam files deduplicated using Picard tools, and GATK best practices were followed for SNP and indel detection (*DePristo et al., 2011*).

## Immunoblotting

Lysates for immunoblotting were prepared from 40 mL cultures grown in 7H9 OADC to an $OD_{600}$ of between 0.5 and 0.8. Cells were cooled to 4°C on ice, centrifuged, and washed 1× with PBS. Washed pellets were then suspended in 0.4 mL of PBS, and ~100 µL of zirconia beads were added. Lysis was performed by 3, 45 s pulses in a BioSpec Mini24 beadbeater with 5 min intervening rest periods on ice. Beads were then removed by centrifugation at 1000×g for 5 min, and the resulting supernatant was mixed 1:1 with 2× Laemmli sample buffer supplemented with .1 M DTT. 20 µL of each sample, heated for 10 min at 100°C, was then separated on 4–12% NuPAGE bis-tris poly acrylamide gels. Separated proteins were then transferred to nitrocellulose and probed with the appropriate antibodies. Antibodies used in this study are monoclonal anti-HA.11 (BioLegend), monoclonal anti-*E. coli* RNA-polymerase β (BioLegend), rabbit polyclonal anti-GFP (Rockland), rabbit polyclonal anti-CarD, and rabbit polyclonal anti-MymT (a kind gift of Dr. Carl Nathan, *Gold et al., 2008*). His tag fusion proteins were detected using HisProbe-HRP (Thermo Fisher).

## Transcriptional profiling

RNA samples for comparative nitric oxide transcriptional profiling were isolated from cells grown and treated as described above, except that nitric oxide treatment was carried out for 5 hr instead

of 3 hr. RNA sequencing was performed as previously reported (*Hubin et al., 2017*). Full RNAseq dataset is given in *Supplementary file 4* and deposited under NCBI BioProject Accession Number PRJNA719428 (https://www.ncbi.nlm.nih.gov/sra/PRJNA719428).

## Recombinant PdtaS/PdtaR purification

Full-length C-terminal 10x His tagged PdtaS and N-terminal 10xHisSmt3 PdtaR fusions were both expressed and purified from Rosetta 2 (DE3) *E. coli* (Millipore Sigma). Cultures, 2–4 L, grown in Luria–Bertani (LB) media to an $OD_{600}$ of 1.0 at 37°C were equilibrated to 30°C for 30 min prior to induction. Protein expression was then induced with 1 mM IPTG, and the cultures were allowed to grow for 3–4 additional hours at 30°C. Cells were then collected by centrifugation, resuspended in lysis/wash buffer (350 mM sodium chloride, 20 mM Tris-HCl pH 8.0, 20 mM imidazole, 1 mM β-mercaptoethanol) at a ratio of 2 mL of buffer to 1 g of wet pellet weight, and stored at −80°C. The following day frozen cell pellets were thawed and disrupted by two passes through a French pressure cell at an internal pressure of 14,000 psi (Sim-Aminco, 40,000 psi cell). Crude lysates were clarified by centrifugation at 20,000×g for 30 min in an SS-34 Rotor (Sorvall), and then passed over TALON metal affinity resin (Takara) equilibrated in wash buffer. Columns were then washed with 20 column volumes of Wash buffer. Bound proteins were eluted with two-column volumes of Elution Buffer (Wash Buffer + 250 mM imidazole).

After elution from the TALON column, PdtaS 10XHis was dialyzed and concentrated to ~2 mg/mL in a dialysis buffer containing 50 mM Tris-HCl pH 8.0, 50 mM sodium chloride, 100 µM DTT with 40% (v/v) glycerol. Eluted 10xHis-Smt3-PdtaR was cleaved from PdtaR by overnight treatment with Ulp1 protease at 4°C. Free PdtaR was then recovered in the flow-through fraction of a second round of TALON resin purification. PdtaR was then dialyzed and stored in the dialysis buffer.

## Histidine kinase autophosphorylation and phosphotransfer activity

For autophosphorylation assay, 5 µM of the purified histidine kinase PdtaS (HK) was incubated for 20 min in autophosphorylation buffer (50 mM Tris-HCl at pH 8.0, 50 mM KCl, 10 mM $MgCl_2$) containing 50 µM ATP and 1 µCi of $\gamma^{32}$P-labeled ATP at 30°C. In the phosphotransfer assay, 10 µM of RR PdtaR was added to the autophosphorylated HK and incubated for time points described in the figure legends. The reaction was terminated by adding 1× SDS-PAGE sample buffer (2% w/v SDS, 50 mM Tris.HCl, pH 6.8, 0.02% w/v bromophenol blue, 1% v/v β-mercaptoethanol, 10% v/v glycerol). The samples were resolved on 15% v/v SDS-PAGE. After electrophoresis, the gel was washed and exposed to phosphor screen (Fujifilm Bas cassette2, Japan) for 4 hr followed by imaging with Typhoon 9210 phosphorimager (GE Healthcare, USA). Quantitative densitometric analysis was performed with ImageJ.

## PdtaS autophosphorylation inhibition assays

5 µM of the purified HK PdtaS, PdtaS$^{C53A}$ mutant and PdtaS$_{kinase\ domain}$ (amino acids 277–501) proteins were preincubated for 10 min in the autophosphorylation buffer with copper chloride and zinc sulfate at various concentrations mentioned in the figure legends and with 1 mM of calcium chloride as a negative control. Similarly, the effect of NO and $H_2S$ was studied by preincubating 5 µM of PdtaS, PdtaS$^{C53A}$PdtaS$^{C57A}$, or PdtaS$_{kinase\ domain}$ protein with spermine NONOate (Cayman Chemical Company, USA) for 40 min and sodium hydrosulfide hydrate (Acros Organics, Belgium) as spontaneous $H_2S$ donor for 30 min at various concentrations specified in the figure legends. The negative control (C) for use of spermine NONOate as NO donor was 1 mM of spermine NONOate allowed to exhaust NO release by overnight incubation at room temperature in Tris.Cl buffer at pH = 6.8. The autophosphorylation reaction was then initiated by adding 50 µM ATP along with 1 µCi of $\gamma^{32}$P-labeled ATP at 30°C for 20 min. The reaction was terminated by adding 1× SDS-PAGE sample buffer. The samples were resolved on 15% v/v SDS-PAGE and analyzed as above. The dose-response curves were plotted and inhibition constant ($K_i$) was obtained by fitting the percentage inhibition to concentration plot using one site-specific binding fit using GraphPad Prism software.

## MST determination of binding affinity

Purified HK PdtaS protein (5 µM) was autophosphorylated in autophosphorylation buffer for 20 min at 30°C using 50 µM of ATP to generate phosphorylated HK PdtaS (PdtaS~P). Phosphotransfer

reaction was carried out at 30°C for 10 min from HK PdtaS~P to fluorescently labeled RR PdtaR protein (50 nM) prelabeled using amine coupling Monolith Protein Labeling kit RED-NHS 2nd generation (Cat# MO-L011, NanoTemper Technologies GmbH) as prescribed in the manufacturer's protocol. Binding assays were performed immediately with in vitro transcribed RNAs *ppe1-5′*, *rv3864*, and *dnaK* at concentrations from 0.31 nM to 10 µM diluted with autophosphorylation buffer having 0.025% Tween-20. The sample was then loaded into standard treated capillaries and analyzed using a Monolith NT. 115 (NanoTemper Technologies GmbH). The red laser was used for a duration of 35 s for excitation (MST power = 40%, LED power 100%). The data were analyzed using MO Control software (NanoTemper Technologies GmbH) to determine the apparent $K_D$ values represented as fraction bound.

For assays using unphosphorylated PdtaR, 50 nM of the fluorescently labeled PdtaR protein was incubated with 50 µM of ATP for 20 min at 30°C in an autophosphorylation buffer. MST assay was immediately performed post addition with *ppe1-5′* (concentration range from 0.15 nM to 10 µM), *rv3864* (concentration range from 0.61 nM to 10 µM), and *dnaK* (concentration range from 0.15 nM to 10 µM) diluted with autophosphorylation buffer having 0.025% Tween-20, and apparent $K_D$ values represented as fraction bound were similarly obtained.

## Acknowledgements

This work was funded by NIH grants R01AI138446, U19AI11143, and P30 CA008748. The authors thank Carl Nathan and Ben Gold for providing the anti-MymT antibody.

## Additional information

### Competing interests

Michael S Glickman: MSG is a consultant for Fimbrion Therapeutics, serves on the SAB of Vedanta Biosciences and receives consulting fees and equity, and serves of the SAB of PRL NYC. The other authors declare that no competing interests exist.

### Funding

| Funder | Grant reference number | Author |
|---|---|---|
| National Institute of Allergy and Infectious Diseases | R01AI138446 | John A Buglino<br>Gaurav D Sankhe<br>Michael S Glickman |
| National Cancer Institute | P30 CA008748 | John A Buglino<br>Gaurav D Sankhe<br>Nathaniel Lazar<br>James M Bean<br>Michael S Glickman |
| National Institute of Allergy and Infectious Diseases | U19AI11143 | John A Buglino<br>James M Bean<br>Michael S Glickman |

The funders had no role in study design, data collection and interpretation, or the decision to submit the work for publication.

### Author contributions

John A Buglino, Conceptualization, Investigation, Visualization, Methodology, Writing - review and editing; Gaurav D Sankhe, Nathaniel Lazar, Investigation, Methodology; James M Bean, Formal analysis, Visualization, Methodology; Michael S Glickman, Conceptualization, Resources, Supervision, Funding acquisition, Investigation, Visualization, Methodology, Writing - original draft, Project administration, Writing - review and editing

## Author ORCIDs

John A Buglino (ID) https://orcid.org/0000-0001-8961-8672
Gaurav D Sankhe (ID) https://orcid.org/0000-0002-5627-2879
James M Bean (ID) https://orcid.org/0000-0001-6733-5794
Michael S Glickman (ID) https://orcid.org/0000-0001-7918-5164

## Ethics

Animal experimentation: Care, housing, and experimentation on laboratory mice were performed in accordance with the National Institute of Heath guidelines, and the approval of the Memorial Sloan-Kettering Institutional Animal Care and Use Committee (IACUC).Protocol approval number is 01-11-030.

## Decision letter and Author response

Decision letter https://doi.org/10.7554/eLife.65351.sa1
Author response https://doi.org/10.7554/eLife.65351.sa2

# Additional files

## Supplementary files

• Supplementary file 1. Bacterial strains used in this work with strain ID, genotype, and relevant features and reference.

• Supplementary file 2. Plasmids used in this work.

• Supplementary file 3. Oligonucleotides used in this work.

• Supplementary file 4. RNAseq count data from RNA sequencing of WT, Δ*rip1*, Δ*pdtaR*, and Δ*rip1*/Δ*pdtaR* exposed to vehicle of diethylenetriamine nitric oxide (DETA-NO) in triplicate. Each cell contains the normalized counts for each gene in each replicate. Please see Materials and methods for details.

• Transparent reporting form

## Data availability

Raw RNA sequencing data has been deposited into SRA, accession number PRJNA719428. The processed data is included in the supplementary data.

The following dataset was generated:

| Author(s) | Year | Dataset title | Dataset URL | Database and Identifier |
|---|---|---|---|---|
| Buglino J, Sankhe G, Lazar N, Bean JM, Glickman MS | 2021 | Rip1 and PdtaR mutant transcriptional responses to Nitric oxide | https://www.ncbi.nlm.nih.gov/sra/PRJNA719428 | NCBI BioProject, PRJNA719428 |

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
