## [Decision Letter]

**Acceptance summary:**

In this study the authors identify several components of a regulatory circuit in *Mycobacterium tuberculosis*, the causative agent of tuberculosis, that allows this important pathogen to respond in a novel manner to nitric oxide and copper ions. The experiments demonstrate that a two-component system, together with a protease, sense and integrate NO stress to control expression of biosynthetic genes required for synthesis of high affinity copper-binding molecules. The authors thus demonstrate a complex set of mechanisms for response to important stress determinants that prevail during colonization of macrophages. The study will be of interest to bacterial physiologists, infectious disease researchers and microbiologists.

**Decision letter after peer review:**

Thank you for submitting your article "Integrated sensing of host stresses by inhibition of a cytoplasmic two component system controls Mtb infection" for consideration by *eLife*. Your article has been reviewed by 3 peer reviewers, and the evaluation has been overseen by Bavesh Kana as the Senior and Reviewing Editor. The following individuals involved in review of your submission have agreed to reveal their identity: Martin Voskuil (Reviewer #2); Jeffery S Cox (Reviewer #3).

The reviewers have discussed the reviews with one another and the Reviewing Editor has drafted this decision to help you prepare a revised submission.

Summary:

In this submission, Buglino and colleagues investigate a regulatory circuit in *Mycobacterium tuberculosis* that facilitates the response to nitric oxide and copper ions, both of which are components of antibacterial response in macrophages. They build on previous work demonstrating that the membrane-embedded site-2 protease (S2P), Rip1 is required for virulence in the mouse model of tuberculosis infection, an effect that is related to activity of σ factors. In this work, the authors further explore the role of this protease in regulating the response to stress conditions.

Key findings:

1. In *M. tuberculosis*, Rip1 is required for resistance to copper ions, nitric oxide, and zinc, with deletion of rip1 resulting in a 10 000 fold increase in sensitivity to copper, which did not appear to result from the activity of any known copper pathways in this organism.

2. In the mouse model of tuberculosis infection, Rip1 appears to defend against the antimicrobial effect of NO, as attenuate virulence of the mutant was reversed in infections of NOS2 deficient mice.

3. Using a screen for suppression of the copper sensitivity, two mutations were detected in PdtaS, suggesting that the PdtaS/R system was involved in the observed response. The authors confirm this by complementing the mutant with variants of PdtaS that lack kinase activity and by creating a deletion mutant of the two-component regulator.

4. The authors then reconstitute the Pdta phosphotransfer reaction and show that this two-component system is able to sense copper, zinc and nitric oxide. Binding of these ligands inhibits signalling through the system. Ligand binding requires the GAF domain of the sensor and required a cysteine residue.

5. Using RNA sequencing of their mutant panel, the authors define the set of genes regulated by the PdtaS/R system, including a gene cluster responsible for the synthesis of isonitrile lipopeptide chalkophores (which bind copper with high affinity) and PPE1-5.

6. To further investigate the link with PPE1-5, the authors demonstrate that PPE1-5 RNA binds to PdtaR and these combined effects are critical for chalkophore biosynthesis and NO resistance in vitro and during infection.

Conclusion: The study shows that the PdtaS/R two-component system and the Rip1 protease sense and integrate NO stress to control expression of biosynthetic genes required for synthesis of high affinity copper-binding molecules.

Essential revisions:

All reviewers commented on the high quality and rigour of the data presented in the manuscript but also raised concerns that there was insufficient evidence presented to support the conclusion that the circuit is important for copper stress resistance. It is our assessment that your work still has significant value without a strong focus on copper stress resistance. I suggest you modify the manuscript to keep the focus on NO stress resistance and indicate the role for copper in regulation of the circuit is not firmly delineated. In a separate section below, I have provided the reviewer suggestions for additional copper experiments to give you a sense of the concerns and how best to address them. You can address these by modifying the manuscript or adding any new supporting data, if these are available. Extensive new experimentation on this aspect is not expected.

Concerns regards copper sensing/resistance:

It seems counterintuitive that a copper acquisition system would be activated during copper stress and it appears that it is probably not from the data presented, although chalkophore gene expression during copper stress was not investigated. If copper is blocking PdtaS/R signaling as is indicated by the in vitro biochemistry, then the unknown Rip1 mediator may be preventing the downstream activation of the chalkophore biosynthesis genes. Although this would be an opposite role for Rip1 than that convincingly demonstrated with NO. Figure 5D shows overexpression of the chalkophore biosynthetic operon instead of increasing copper resistance, as is the case with NO, results in a modest increase in sensitivity to copper stress on plates. However, deletion of the nrp gene, which is part of the chalkophore biosynthetic operon, appears to also modestly increase sensitivity to copper stress on plates (Figure S7B and C). These two findings provide the phenotypic data that suggest a role for the system in copper stress resistance, but they also appear to be contradictory findings. In addition, restoration of chalkophore gene expression or expression of PPE5' or PPE1 did not rescue copper sensitivity of the rip1 mutant strains demonstrating that chalkophore regulation is not likely the mechanism in which loss of Rip1 causes copper sensitivity. Since the PdtaS/R system controls chalkophore synthesis it seems unlikely that the in vitro finding that copper blocks the system like NO occurs in the cell during copper stress. Alternatively, if copper mediate blockage of activation does occur in the bacilli, it may not result in de-repression of chalkophore biosynthesis genes as occurs with NO. As gene expression was not determined during copper stress, the impact of copper on gene expression cannot be assessed. While it appears that copper stress is in some manner integrated into the sensory inputs of the overall circuit, the above findings make the role of copper sensing vague. The authors indicate that the PdtaS/R system is not involved in copper resistance, but it is not clear why copper would not trigger de-repression of chalkophore genes as it appears to block the system like NO. To establish the role of copper in the system several experiments would be helpful. These include expression profiling during copper stress to determine if genes such as the chalkophore operon are regulated during copper stress as they are induced by NO.

Copper stress experiments could also be performed in liquid culture to determine true lethality upon plating instead of copper exposure only on plates. As copper challenge experiments were done with copper added to plates not in liquid culture, as was the case for NO and zinc experiments, it is difficult to compare NO and copper experiments. As solid agar copper toxicity experiments cannot differentiate lethal and static effects of copper, the copper experiments should be labeled as "relative inhibition" instead of "relative survival."

The Ki for copper binding PdtaS was found to be 34 microM, since PdtaS is cytosolic and free intracellular copper is known to be highly toxic, this level of intracellular copper should be put into context with potential intracellular levels.

A mechanism should be discussed to incorporate the contradictory findings that copper should activate chalkophore synthesis genes through its interaction with PdtaS, even though this interaction seems to not have a role. It may be the impact on PdtaS autophosphorylation by copper is unavoidable due to the chemistry therefor the Rip1 system is employed to short circuit the activation of chalkophore synthesis from copper stress to prevent increased copper accumulation.

The paragraph starting on line 103: The logical basis for the epistasis tests is not entirely clear. The first idea makes sense – Rip1 mutants may be sensitive to high copper levels because they are unable to derepress the RicR and CsoR regulons which function to export copper, and thus removal of the repressors would suppress the sensitivity phenotype of the rip1 mutant. While the rip1 csoR double mutant is sensitive to high copper, isn't it possible that RicR is still repressing something that is required for copper resistance. That is, wouldn't it be more definitive to remove both ricR and csoR? This needs some clarification. Secondly, the transition from the two copper regulators (RicR and CsoR) to the two gene targets is not intuitive, it would help the reader if the functions of MymT and CtpV are explained. This part is a bit confusing as the mymT and ctpV mutants aren't sensitive to copper concentrations that are toxic to rip1 mutant cells – isn't that the main evidence that the Rip1 pathway for copper resistance is mediated through mechanisms other than MymT or CtpV mediated copper efflux (or perhaps not due to efflux at all)? Or are the mymT and ctpV mutants still able to efflux copper? Here the logic for the epistasis test isn't clear – it would seem that if the Rip1 pathway for copper resistance is independent of the MymT/CtpV pathway, the main expectation of double mutants would be that there would be ultra-sensitivity, i.e. synergy or "negative epistasis". It seems that there is a fairly potent synergistic effect of ctpV and csoR in combination with the rip1 mutation in Figure S3C, especially at medium [Cu] (2 and 3 logs). This is not evident with mymT. Finally, the term "copper handling" is vague (line 116).

Major concerns that must be addressed:

1. In order for the authors to claim that NO and copper modulates the PtdaS/R/Rip1 signal transduction system via direct interaction with a thiol switch, they need to demonstrate a direct effect of copper or NO on the Cys residue. For example, evidence of S-nitrosylation, the covalent addition of NO moiety to the sulfur atom of Cys, would provide conclusive evidence of a thiol switch. Also, the authors have not demonstrated whether copper could modify the Cys resideu. Standard spectrophotometry, mass spectrometry or other assays (e.g., biotin switch) could demonstrate these modifications. Also, if the Cys residue plays a key role, why was not it identified in the suppressor screen? What about the 2nd Cys residue (C57)? In fact, other residues were identified strongly suggesting that Cys is not central to a thiol switch as other residues (e.g., V) are also involved. As indicated above, rephrasing these conclusions and removing the emphasis on copper should assist in addressing these issues.

2. Did the authors consider that PdtaS may bind other cofactors (e.g. c/gAMP) or prosthetic groups such as heme? Although doi.org/10.1016/j.jsb.2011.11.012 reported that there is no evidence of a bound metabolite they did not comment on the presence of heme in the GAF domain and stated that the internal cavity is too small to bind heme in the PAS domain. Although not required, since the authors have purified PdtaS, a simple UV-vis scan will reveal whether this protein binds heme (potentially loosely as it was not identified in the crystal structure), which will provide a clear mechanism of responding to NO and potentially copper.

3. The rational for performing RNA seq experiments using NO and not also copper is unclear. Can you explain this reasoning? The main reason for this request is you identified a putative copper binding operon under NO stress. Please also expand on your statement that isonitrile binds copper with high affinity as this binding data is not provided in the references (at least not in the PNAS paper), which suggests it is involved in copper transport and not binding. Strong copper binding is also mentioned in lines 303 without any reference; please provide.

4. In figure 6, the authors examined 4 mutants (some of them are double knockouts) in vivo, but no complementation experiments were performed. We understand the challenges and complexities involved in re-doing these time-consuming experiments however, genetic complementation is the standard in the field. Can the authors provide a compelling reason for the lack of complementation?

5. Although the authors predict in lines 323-326 that the in vivo virulence defect conferred by loss of rip1 should be reversed in a rip1/pdtaR knockout, this is not entirely true. Although the authors attempt to explain this by separating the response during acute infection versus chronic infection, this is not entirely consistent with their prediction in lines 323 – 326 and the data points in figure 6C and D. This will become more evident if the authors provide the statistics for not only the wild-type and rip1 comparison, but also for the wild-type and rip1/pdtaR and wild-type and pdtaR comparisons. As it stands, it is incomplete.

6. Lines 133-134. this conclusion is too strong – it seems that increased growth of an Mtb mutant in an iNOS knockout mouse does not establish that the gene is directly involved in defending against the antimicrobial effect of NO. Certainly this is consistent with the direct NO sensitivity of the mutant in culture, but there are many effects of NO during anti-TB immune responses.

7. Figure 3. There is a bit of a disconnect between the gel in panel B and the quantification in panel F. Perhaps it is the pdf but while there is a noticeable decrease in the band intensity in the two-fold drop from 5 to 10 uM, but the 10-100 μm bands all look the same to me (unlike the Zn inhibition data in Figure S5, which looks really good). Also, a -Cu control is missing – how was the % inhibition calculated without this?

8. Isn't the simple expectation that chalkophore synthesis would lead to Cu resistance? It seems unexpected that it would only affect NO and not Cu sensitivity. That is mentioned in the Discussion section but it seems confusing in the results without any explanation.

9. The role of PPE1-5' RNA could be more thoroughly explored. First, the expression of the PPE1-5' RNA is in the context of expression of the entire PPE1 gene. Thus, it is not clear that expression of PPE1-5' is sufficient to suppress the NO sensitivity phenotype of the Rip1 mutant. Can just the RNA be expressed?

---

## [Author Response]

Essential revisions:All reviewers commented on the high quality and rigour of the data presented in the manuscript but also raised concerns that there was insufficient evidence presented to support the conclusion that the circuit is important for copper stress resistance. It is our assessment that your work still has significant value without a strong focus on copper stress resistance. I suggest you modify the manuscript to keep the focus on NO stress resistance and indicate the role for copper in regulation of the circuit is not firmly delineated. In a separate section below, I have provided the reviewer suggestions for additional copper experiments to give you a sense of the concerns and how best to address them. You can address these by modifying the manuscript or adding any new supporting data, if these are available. Extensive new experimentation on this aspect is not expected.Concerns regards copper sensing/resistance:It seems counterintuitive that a copper acquisition system would be activated during copper stress and it appears that it is probably not from the data presented, although chalkophore gene expression during copper stress was not investigated. If copper is blocking PdtaS/R signaling as is indicated by the in vitro biochemistry, then the unknown Rip1 mediator may be preventing the downstream activation of the chalkophore biosynthesis genes. Although this would be an opposite role for Rip1 than that convincingly demonstrated with NO. Figure 5D shows overexpression of the chalkophore biosynthetic operon instead of increasing copper resistance, as is the case with NO, results in a modest increase in sensitivity to copper stress on plates. However, deletion of the nrp gene, which is part of the chalkophore biosynthetic operon, appears to also modestly increase sensitivity to copper stress on plates (Figure S7B and C). These two findings provide the phenotypic data that suggest a role for the system in copper stress resistance, but they also appear to be contradictory findings. In addition, restoration of chalkophore gene expression or expression of PPE5' or PPE1 did not rescue copper sensitivity of the rip1 mutant strains demonstrating that chalkophore regulation is not likely the mechanism in which loss of Rip1 causes copper sensitivity. To establish the role of copper in the system several experiments would be helpful. These include expression profiling during copper stress to determine if genes such as the chalkophore operon are regulated during copper stress as they are induced by NO.

We agree with the reviewers that some further explanation is required, and that our study does not delineate the full mechanisms of Cu resistance downstream of Rip1. However, we disagree that the findings are contradictory. The statement that “ a copper acquisition system would be activated during copper stress” seems to assume that the *nrp* operon is involved in copper acquisition because the natural product produced by these enzymes binds copper. The molecular function of the molecules produced by the *nrp* operon encoded proteins is not known. The reviewers seem to assume that any molecule that binds copper is acting to modulate copper toxicity via binding, as is the case for copper binding proteins such as MymT. However, in the case of the chalkophores of Methanococcus, the function of the chalkophore is to supply copper to a membrane bound copper enzyme, not to directly ameliorate toxicity. Although we agree that it is surprising that the chromosomal target of the cascade that mediates NO resistance, but not copper resistance, is a copper binding molecule, the experiments in the paper were executed to definitively demonstrate this point.

Therefore, we agree that our study does not fully define the Cu arm of the pathway. However, what is definitive in our data, and acknowledged in the review, is: (1) Rip1 is required for resistance to both stresses (2) loss of PdtaS/R function is a suppressor of BOTH the NO and Cu sensitivity of the D*rip1* strain, and (3) Cu and NO both inhibit the signaling system biochemically. The fact that the chromosomal targets that mediate resistance to each of these stresses are distinct indicates that there is some mechanism that specifies PdtaR function to specific gene sets under specific stress conditions. Our data identifies the Ppe1-5’ RNA as that factor under NO stress and we hypothesize, but do not identify, a Cu induced similar factor. The fact that Cu and NO both inhibit PdtaS must be understood in the context of our two signal model, in which inhibition of the kinase is not sufficient to depress gene expression without a second signal (PPE1-5’ in the case of NO) and therefore the assertion in the review that inhibition of PdtaS by copper would stimulate *nrp* expression is not correct. The RNAseq data in the PdtaR ko in the absence of any stress demonstrates this unambiguously, there is no induction of the *nrp* operon in the PdtaR ko in basal conditions.

In addition, the review points to several pieces of data for example “Figure 5D shows…” and “appears to also modestly increase sensitivity to copper stress.” that are nonsignificant differences and therefore were not highlighted in our analysis or integrated into the model. They should not be invoked as supporting the assertion that our data is “contradictory”.

In summary, although we agree that our study does not resolve the ultimate downstream chromosomal target of the Rip1/PdtaS/PdtaR cascade that controls Cu resistance, we do definitively implicate this cascade in Cu resistance. The fact that a Cu binding molecule mediates NO, but not Cu resistance, may be surprising based on the notion that any Cu binding molecule must be ameliorating toxicity, but this is a presumption that is not borne out in the data. Multiple other hypotheses can be advanced but must be explored by further experimentation. For example, NO may induce a Cu deficient state in membrane bound enzymes which is ameliorated by a restorative function of chalkophore-bound Cu through a loading function.

In response to these points raised by the reviewers, we have clarified the discussion to better incorporate these concerns and make clear that our study does not elucidate the downstream mechanisms of Cu resistance controlled by the Rip1/PdtaS/R system. We have focused the data presentation more on the NO phenotypes. We have maintained the model figure to emphasize this point because this visual representation clearly indicates a “?” on the chromosomal target of Cu arm of the pathway.

Copper stress experiments could also be performed in liquid culture to determine true lethality upon plating instead of copper exposure only on plates. As copper challenge experiments were done with copper added to plates not in liquid culture, as was the case for NO and zinc experiments, it is difficult to compare NO and copper experiments. As solid agar copper toxicity experiments cannot differentiate lethal and static effects of copper, the copper experiments should be labeled as "relative inhibition" instead of "relative survival."

We thank the reviewer for this suggestion. We have added data to Figure 1-Figure Supplement 1D showing that the copper sensitivity of the Δ*rip1* strain is present when cells are exposed to Cu in liquid media.

The Ki for copper binding PdtaS was found to be 34 microM, since PdtaS is cytosolic and free intracellular copper is known to be highly toxic, this level of intracellular copper should be put into context with potential intracellular levels.

Thank you for this question. The intracellular copper concentration noted in phagosomes is in the range of 20 – 40 μm (Wagner D, et al. J Immunol 2005;174:1491–1500). It is difficult extrapolate these levels to the physiologic relevance to PdtaS particularly because of the equilibrium between Cu^+1^ and Cu^+2^. Although we apply Cu^+2^ in these assays, we cannot control the conversion to Cu^+1.^ As a point of reference, the biochemical data in PMID 21166899 indicates that dissociation of RicR from its operator requires 15-20 μM copper, suggesting that the cytosolic concentration reaches this level under copper stress given the robust induction of RicR targets under the conditions we test.

Since the PdtaS/R system controls chalkophore synthesis it seems unlikely that the in vitro finding that copper blocks the system like NO occurs in the cell during copper stress. Alternatively, if copper mediate blockage of activation does occur in the bacilli, it may not result in de-repression of chalkophore biosynthesis genes as occurs with NO. As gene expression was not determined during copper stress, the impact of copper on gene expression cannot be assessed. While it appears that copper stress is in some manner integrated into the sensory inputs of the overall circuit, the above findings make the role of copper sensing vague. The authors indicate that the PdtaS/R system is not involved in copper resistance, but it is not clear why copper would not trigger de-repression of chalkophore genes as it appears to block the system like NO.A mechanism should be discussed to incorporate the contradictory findings that copper should activate chalkophore synthesis genes through its interaction with PdtaS, even though this interaction seems to not have a role. It may be the impact on PdtaS autophosphorylation by copper is unavoidable due to the chemistry therefor the Rip1 system is employed to short circuit the activation of chalkophore synthesis from copper stress to prevent increased copper accumulation.

As noted above, this comment presumes the inhibition of PdtaS/R alone is sufficient to activate all downstream gene expression, which is not correct. We have performed RNA sequencing of the PdtaR mutant without stress and find very few differences in gene expression (see figure 4B,C) and specifically that the chalkophore operon is not induced. This piece of data is part of the support for our two-signal model in which inhibition of PdtaS signaling is necessary but not sufficient for activation of gene expression. Our data indicate that a second signal to modify PdtaR binding is required, and, in the case of NO, we identify this signal as the NO induced transcription of PPE1-5’. The reviewer is correct that we have not identified this second signal for Cu, nor identified the chromosomal target(s) it regulates, but the data does not support the assertion that Cu inhibition of PdtaS is sufficient to activate chalkophore biosynthesis and that, based this is assumption, our findings are contradictory.

The paragraph starting on line 103: The logical basis for the epistasis tests is not entirely clear. The first idea makes sense – Rip1 mutants may be sensitive to high copper levels because they are unable to derepress the RicR and CsoR regulons which function to export copper, and thus removal of the repressors would suppress the sensitivity phenotype of the rip1 mutant. While the rip1 csoR double mutant is sensitive to high copper, isn't it possible that RicR is still repressing something that is required for copper resistance. That is, wouldn't it be more definitive to remove both ricR and csoR? This needs some clarification.

The RicR and CsoR regulons are nonoverlapping according to prior literature. It is possible there is redundancy in function, but the experiment requested would require creation of triple mutant and based on the data that all RicR targets measured are intact, in addition to the lack of epistasis with RicR targets (i.e. CtpV and MymT, see below), we have not created this mutant.

Secondly, the transition from the two copper regulators (RicR and CsoR) to the two gene targets is not intuitive, it would help the reader if the functions of MymT and CtpV are explained. This part is a bit confusing as the mymT and ctpV mutants aren't sensitive to copper concentrations that are toxic to rip1 mutant cells – isn't that the main evidence that the Rip1 pathway for copper resistance is mediated through mechanisms other than MymT or CtpV mediated copper efflux (or perhaps not due to efflux at all)? Or are the mymT and ctpV mutants still able to efflux copper? Here the logic for the epistasis test isn't clear – it would seem that if the Rip1 pathway for copper resistance is independent of the MymT/CtpV pathway, the main expectation of double mutants would be that there would be ultra-sensitivity, i.e. synergy or "negative epistasis". It seems that there is a fairly potent synergistic effect of ctpV and csoR in combination with the rip1 mutation in Figure S3C, especially at medium [Cu] (2 and 3 logs). This is not evident with mymT.

The reviewer is correct, the data shows that the Cu resistance phenotype of the Rip1 strain is independent of MymT and CtpV. The confusion may arise because the MymT and CtpV mutants are not themselves sensitive to Cu, but this result is actually consistent with the published literature, which shows variable copper sensitivity in *M. tuberculosis* strains. For example, CDC1551 (more similar to Erdman than H37Rv) is 100 fold more resistant to Cu than H37Rv, and loss of *mmcO*, which confers severe sensitivity to Cu in H37Rv, has no phenotype in CDC1551 (PMID 24549843). With regard to epistasis, if the CtpV or MymT mutants are not sensitive to Cu, but the *rip1* mutant is, then it is not plausible that the phenotype of the Rip1 strain is due to MymT or CtpV dysfunction. If the CtpV or MymT mutants were indeed sensitive, then the predicted result would have been additive sensitivity, i.e. lack of epistasis.

Finally, the term "copper handling" is vague (line 116).

We have modified this wording.

Major concerns that must be addressed:1. In order for the authors to claim that NO and copper modulates the PtdaS/R/Rip1 signal transduction system via direct interaction with a thiol switch, they need to demonstrate a direct effect of copper or NO on the Cys residue. For example, evidence of S-nitrosylation, the covalent addition of NO moiety to the sulfur atom of Cys, would provide conclusive evidence of a thiol switch. Also, the authors have not demonstrated whether copper could modify the Cys resideu. Standard spectrophotometry, mass spectrometry or other assays (e.g., biotin switch) could demonstrate these modifications.

Our apologies, our intention in using the term “thiol switch” was not to imply that the PdtaS cysteine is directly modified by either Cu or NO, which we did not show, and we were unaware that the term “thiol switch” implicitly indicates covalent modification. Our intended meaning was that the thiol (originally C53 but see below for our new data also implicating C57) is required for the NO and Cu inhibition of PdtaS kinase activity. The structural and biochemical mechanism of this thiol dependent inhibition will require further study, but our data does demonstrate that PdtaS lacking either thiol cannot be inhibited by NO or Cu. We have altered our terminology to eliminate the term “thiol switch.”

Also, if the Cys residue plays a key role, why was not it identified in the suppressor screen?

There could be multiple explanations for this finding such as non-saturation of the screen, some maladaptive property of the Cysteine mutation in vivo that prevents its isolation, or the fact that both thiols seem to be involved in sensing and therefore a mutation that lies in proximity to both may uniquely disrupt their geometry in a way that is maximally suppressive.

What about the 2nd Cys residue (C57)? In fact, other residues were identified strongly suggesting that Cys is not central to a thiol switch as other residues (e.g., V) are also involved. As indicated above, rephrasing these conclusions and removing the emphasis on copper should assist in addressing these issues.

We thank the reviewer for this question. We agree that C57 is plausibly involved due to its proximity to C53 and to the suppressor mutation that was identified. To investigate this possibility, we purified PdtaS C57A and assayed this protein for kinase activity and inhibition by Cu and NO. We found that the C57A mutation did not modify the basal activity of the kinase but did confer resistance to inhibition by the NO (see revised Figure 3G and Figure 3-Figure Supplement 2). We were unable to assay this protein for Cu inhibition due to solubility problems in the assay with high Cu concentrations and therefore the role of this residue in Cu inhibition remains unresolved. Figure 7 has been updated to incorporate C57.

2. Did the authors consider that PdtaS may bind other cofactors (e.g. c/gAMP) or prosthetic groups such as heme? Although doi.org/10.1016/j.jsb.2011.11.012 reported that there is no evidence of a bound metabolite they did not comment on the presence of heme in the GAF domain and stated that the internal cavity is too small to bind heme in the PAS domain. Although not required, since the authors have purified PdtaS, a simple UV-vis scan will reveal whether this protein binds heme (potentially loosely as it was not identified in the crystal structure), which will provide a clear mechanism of responding to NO and potentially copper.

We thank the reviewer for this question, which we have also considered. We are aware of reports that the PdtaS binds cyclic-di-GMP (https://www.biorxiv.org/content/10.1101/615575v1.abstract) at the GAF domain of the sensor. The interaction of this metabolite with Cu and NO sensing documented here is an interesting subject for further study but beyond the scope of this study. It does suggest that PdtaS integrates multiple inputs and we have added this point to the discussion.

With regard to heme binding by PdtaS, we have performed UV-vis spectroscopy on our purified protein in comparison to BSA as a negative control and cytochrome C as a positive control. We see clear absorbance for CytC, but no signal for either BSA or PdtaS, suggesting that PdtaS does not contain heme, as previously reported. The graph is shown in Author response image 1.

**Author response image 1. sa2fig1:** 

3. The rational for performing RNA seq experiments using NO and not also copper is unclear. Can you explain this reasoning?

We have added this rationale to the text:

“We focused on NO stress for several reasons, including: (1) the strong phenotypic reversion of D*rip1* in iNOS deficient mice (2) the strong phenotypic reversion of Rip1 by the *pdtaR* mutation and (3) the technical advantages of treating Mtb with carefully titrated NO donors in liquid culture. “

The main reason for this request is you identified a putative copper binding operon under NO stress. Please also expand on your statement that isonitrile binds copper with high affinity as this binding data is not provided in the references (at least not in the PNAS paper), which suggests it is involved in copper transport and not binding. Strong copper binding is also mentioned in lines 303 without any reference; please provide.

The copper binding properties of the isonitrile chalkophores is reported in reference 42 of the manuscript (Wang et al. ACS Chemical biology) and in reference 46 (Xu et al). Reference 46 uses synthetic molecules of defined structure from *Streptomyces*. Neither the PNAS paper, nor the ACS chemical biology paper (reference 42) define the physiologic role of the molecules, only that they bind Cu.

4. In figure 6, the authors examined 4 mutants (some of them are double knockouts) in vivo, but no complementation experiments were performed. We understand the challenges and complexities involved in re-doing these time-consuming experiments however, genetic complementation is the standard in the field. Can the authors provide a compelling reason for the lack of complementation?

We agree with the reviewer that complementation is an essential genetic control. We have included complementation in Figure 6A,B for the Δ*nrp* strain. For the Δ*rip1* strain in figure 6C,D, the complemented strain has been previously reported in mice and is therefore not repeated here (although it is included in all in vitro assays, see all prior figures). For the *rip1*/*pdtaR* double mutant, we are interrogating reversion of the Rip1 in vivo phenotype by the PdtaR null mutation. The complementation control we assume the reviewer is asking about is testing is the animal phenotype of Δ*rip1*Δ*pdtaR*+*pdtaR* to show that the reversion we observe is due to the *pdtaR* mutation and not a spontaneous suppressor. We acknowledge that we have not done this infection due to the large number of strains, time, and expense, but note that this strain is included in every in vitro assay in Figure 2, alongside analogous complemented strains for the PdtaS suppressors. Both the PdtaS SNP, the Δ*pdtaS::hyg* mutation, and the Δ*pdtaR::hyg* mutation all share the same reversion phenotype, strongly arguing against a spontaneous mutation. In addition, in all assays in Figure 2, these suppressors are complemented, and, in all cases, the suppressor effect is reverted by complementation.

5. Although the authors predict in lines 323-326 that the in vivo virulence defect conferred by loss of rip1 should be reversed in a rip1/pdtaR knockout, this is not entirely true. Although the authors attempt to explain this by separating the response during acute infection versus chronic infection, this is not entirely consistent with their prediction in lines 323 – 326 and the data points in figure 6C and D. This will become more evident if the authors provide the statistics for not only the wild-type and rip1 comparison, but also for the wild-type and rip1/pdtaR and wild-type and pdtaR comparisons. As it stands, it is incomplete.

We apologize if there was some display problem with the figure, since the statistics requested are mostly displayed on the figure. We have added additional statistics. We agree with the reviewer that the suppression of the Rip1 virulence phenotype is limited to early infection in the lung, but in this period the suppression is very strong, reversing the Rip1 CFU decrement by approximately 100 fold. The PdtaR single mutant has a virulence phenotype alone, although this was not our focus. However, this result indicates that the suppression of the Rip1 phenotype by PdtaR would not be expected to replicate WT titers since the Rip1 independent virulence function of PdtaR, which we do not define, is still defective in the Rip1 PdtaR double mutant. Indeed, there is a significant difference between WT and PdtaR and WT vs Rip1/PdtaR for this reason (see added stats in Figure 6). We do not believe the stage specific reversion of the Rip1 phenotype in the lung is a weakness; it shows that the signal transduction cascade we identify is only relevant in early infection.

6. Lines 133-134. this conclusion is too strong – it seems that increased growth of an Mtb mutant in an iNOS knockout mouse does not establish that the gene is directly involved in defending against the antimicrobial effect of NO. Certainly this is consistent with the direct NO sensitivity of the mutant in culture, but there are many effects of NO during anti-TB immune responses.

We have modified this wording to accommodate this possibility.

7. Figure 3. There is a bit of a disconnect between the gel in panel B and the quantification in panel F. Perhaps it is the pdf but while there is a noticeable decrease in the band intensity in the two-fold drop from 5 to 10 uM, but the 10-100 μm bands all look the same to me (unlike the Zn inhibition data in Figure S5, which looks really good). Also, a -Cu control is missing – how was the % inhibition calculated without this?

The densitometric analysis of the band displayed in percentage inhibition dose response plots are mean of replicates while the image is representative of it. Instead of a no copper control, we have used an unrelated divalent cation, Ca^2+^, as a negative control to control for charge. Calculations are made in relation to this control.

8. Isn't the simple expectation that chalkophore synthesis would lead to Cu resistance? It seems unexpected that it would only affect NO and not Cu sensitivity. That is mentioned in the Discussion section but it seems confusing in the results without any explanation.

See above. It was our expectation as well, but the data does not support this idea. In this case, the simplest model, one of copper chelation by copper binding, is not the correct model. We have modified the discussion to expand on this point.

9. The role of PPE1-5' RNA could be more thoroughly explored. First, the expression of the PPE1-5' RNA is in the context of expression of the entire PPE1 gene. Thus, it is not clear that expression of PPE1-5' is sufficient to suppress the NO sensitivity phenotype of the Rip1 mutant. Can just the RNA be expressed?

We have not yet been able to define the minimal PPE-5’ element that can reproduce the restoration of NO resistance in vivo. The in vitro biochemistry showing that PdtaR binds directly to the PPE1 5’ RNA uses the RNA (shown in Figure S6A) derived from the PPE1 5’ peak, strengthening our assertion that this RNA is the functionally important element, but we have not yet been able to reproduce this effect in vivo and it will require further experimentation.